# The *Candida albicans* virulence factor candidalysin polymerizes in solution to form membrane pores and damage epithelial cells

Charles M Russell[1†], Katherine G Schaefer[2†], Andrew Dixson[1], Amber LH Gray[3], Robert J Pyron[1], Daiane S Alves[1], Nicholas Moore[1], Elizabeth A Conley[2], Ryan J Schuck[1], Tommi A White[4,5], Thanh D Do[3], Gavin M King[2,4*], Francisco N Barrera[1*]

[1]Department of Biochemistry & Cellular and Molecular Biology, University of Tennessee, Knoxville, United States; [2]Department of Physics and Astronomy, University of Missouri, Columbia, United States; [3]Department of Chemistry, University of Tennessee, Knoxville, United States; [4]Department of Biochemistry, University of Missouri, Columbia, United States; [5]Electron Microscopy Core, University of Missouri, Columbia, United States

*For correspondence:
kinggm@missouri.edu (GMK);
fbarrera@utk.edu (FNB)

[†]These authors contributed equally to this work

Competing interest: The authors declare that no competing interests exist.

**Abstract** *Candida albicans* causes severe invasive candidiasis. *C. albicans* infection requires the virulence factor candidalysin (CL) which damages target cell membranes. However, the mechanism that CL uses to permeabilize membranes is unclear. We reveal that CL forms membrane pores using a unique mechanism. Unexpectedly, CL readily assembled into polymers in solution. We propose that the basic structural unit in polymer formation is a CL oligomer, which is sequentially added into a string configuration that can close into a loop. CL loops appear to spontaneously insert into the membrane to become pores. A CL mutation (G4W) inhibited the formation of polymers in solution and prevented pore formation in synthetic lipid systems. Epithelial cell studies showed that G4W CL failed to activate the danger response pathway, a hallmark of the pathogenic effect of CL. These results indicate that CL polymerization in solution is a necessary step for the damage of cellular membranes. Analysis of CL pores by atomic force microscopy revealed co-existence of simple depressions and more complex pores, which are likely formed by CL assembled in an alternate oligomer orientation. We propose that this structural rearrangement represents a maturation mechanism that stabilizes pore formation to achieve more robust cellular damage. To summarize, CL uses a previously unknown mechanism to damage membranes, whereby pre-assembly of CL loops in solution leads to formation of membrane pores. Our investigation not only unravels a new paradigm for the formation of membrane pores, but additionally identifies CL polymerization as a novel therapeutic target to treat candidiasis.

## Editor's evaluation

This study by Russell et al. reveals interesting details of the mechanism of pore formation of a newly identified peptide toxin secreted by *Candida albicans* (candidalysin). Using atomic force microscopy along with other techniques, the authors demonstrate that pre-assembly of polymers in solution is the first step in the formation of permeabilizing pores, and identify an intriguing inactive mutant. This manuscript will be of interest to several fields, in particular to microbiologists and structural biologists studying pore forming proteins and peptides.

**eLife digest** The fungus *Candida albicans* is the most common cause of yeast infections in humans. Like many other disease-causing microbes, it releases several virulent proteins that invade and damage human cells. This includes the peptide candidalysin which has been shown to be crucial for infection.

Human cells are surrounded by a protective membrane that separates their interior from their external environment. Previous work showed that candidalysin damages the cell membrane to promote infection. However, how candidalysin does this remained unclear.

Similar peptides and proteins cause harm by inserting themselves into the membrane and then grouping together to form a ring. This creates a hole, or 'pore', that weakens the membrane and allows other molecules into the cell's interior. Here, Russell, Schaefer et al. show that candidalysin uses a unique pore forming mechanism to impair the membrane of human cells.

A combination of biophysical and cell biology techniques revealed that the peptide groups together to form a chain. This chain of candidalysin proteins then closes in on itself to create a loop structure that can insert into the membrane to form a pore. Once embedded within the membrane, the proteins within the loops rearrange again to make the pores more stable so they can cause greater damage.

This type of pore formation has not been observed before, and may open up new avenues of research. For instance, researchers could use this information to develop inhibitors that stop candidalysin from forming chains and harming the membranes of cells. This could help treat the infections caused by *C. albicans*.

## Introduction

Fungal infections are responsible for high morbidity burden around the world (*Brown et al., 2012*). *Candida albicans* is one of the most serious fungal threats to human health. This pathogen is part of the commensal flora, but can infect the skin, mouth, vagina, and gut, both in healthy and immunocompromised individuals. Additionally, *C. albicans* causes invasive candidiasis, an infection of the blood, heart, and other organs, which is common in hospitalized patients (*Kullberg and Arendrup, 2015*). Invasive candidiasis causes high rates of mortality (~50%), even when patients are treated with antifungal therapy (*Kullberg and Arendrup, 2015*).

It has been recently discovered that *C. albicans* causes toxicity by secreting onto the surface of epithelial cells a peptide toxin named candidalysin (CL) (*Moyes et al., 2016*). This virulence factor binds to the plasma membrane of target human epithelial cells and compromises the permeability barrier, causing uncontrolled $Ca^{2+}$ influx and release of cellular proteins on to the extracellular medium (*Naglik et al., 2019*). The resulting cellular damage activates a signaling cascade that triggers the release of pro-inflammatory mediators (*Naglik et al., 2019*; *Richardson et al., 2018b*), which participate in the immune response to *C. albicans* infection. Furthermore, CL plays a second pathogenic role, as it also attacks the membrane of mononuclear phagocytes which would otherwise fight the fungal infection (*Kasper et al., 2018*). CL is necessary for infection by *C. albicans* (*Liu et al., 2021*). Therefore, pharmacological inhibition of CL activity is a promising avenue to fight infection of this human pathogen. However, the development of drugs to counter the action of CL is hampered by the lack of understanding of how this peptide damages membranes.

Here, we describe how the use of biophysical techniques (mass photometry, native mass spectrometry, analytical ultracentrifugation, transmission electron microscopy, atomic force microscopy, oriented circular dichroism, and liposome dye release assay), modeling, and cellular assays allowed us to elucidate that CL damages membranes by forming pores that are assembled through a unique molecular mechanism.

## Results

### CL assembles into polymers in solution

When we ran CL on an SDS-PAGE, we observed not only the expected monomeric band running at ~4 kDa, but also a band of slower electrophoretic mobility (*Figure 1A*). This result suggests that

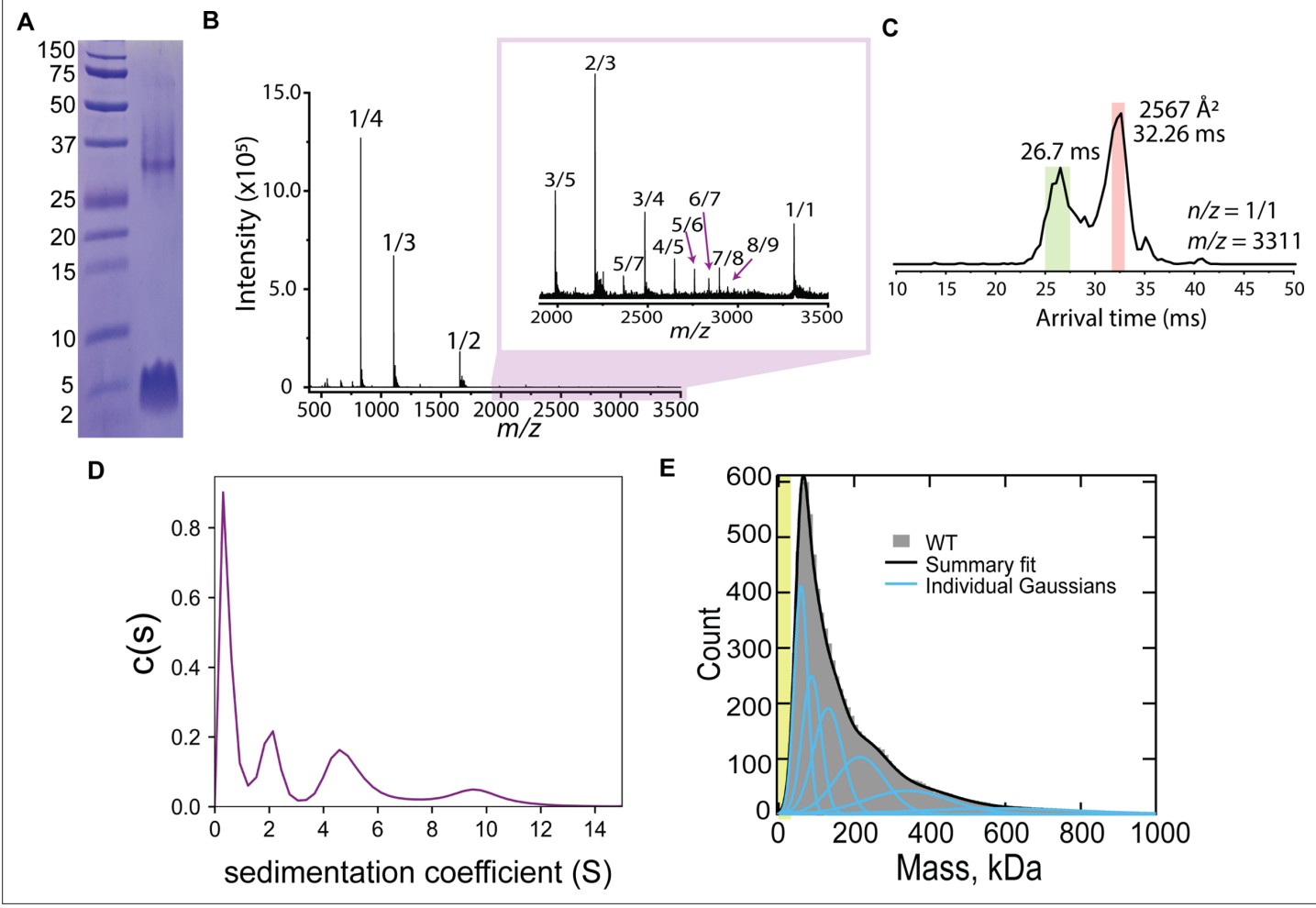

**Figure 1.** CL forms complex assemblies in solution. (**A**) SDS-PAGE of CL shows a~4 kDa monomeric band, and a band corresponding to a large oligomer. Molecular weight markers are shown on the left. (**B**) IM-MS mass spectrum of CL annotated with oligomer-to-charge (n/z) ratios. The inset spectrum reveals oligomers that are not immediately identifiable in the top spectrum. (**C**) Arrival time distribution of the 8-mer (n/z=1/1) species. This peak was more populated than those corresponding to smaller oligomers, which might result from decomposition of the 8-mer in the gas phase. The experimental collisional cross section for the single 8-mer (highlighted in salmon) is given. (**D**) Analytical ultracentrifugation data identifies populations of increasingly larger CL oligomers. (**E**) Mass photometry data of CL oligomeric species. The green area marks the approximate mass range (<50 kDa) below the resolution of the technique. Data distribution was best fit with six Gaussian populations (shown as blue lines, and summary fit is the black line). The mass of the peaks agree with the expected mass for 2, 3, 4, 8, 13, and 22 CL 8-mers (similar to a Fibonacci sequence).

The online version of this article includes the following source data and figure supplement(s) for figure 1:

**Source data 1.** Original SDS-PAGE gel corresponding to *Figure 1A*.

**Source data 2.** Raw mass photometry data corresponding to *Figure 1E*.

**Figure supplement 1.** IM-MS results.

**Figure supplement 2.** Deconvolution of MP data.

CL self-assembles in solution, and we hypothesized that this event might be related to its ability to damage cellular membranes. We therefore investigated CL oligomerization, first using native ion mobility-mass spectrometry (IM-MS), a sensitive analytical technique that can isolate and characterize transient oligomers based on their mass-to-charge ratio (*m/z*), shape, size, and charge. IM-MS analysis of CL revealed several oligomeric species (*Figure 1B*). The mass spectral peak at *m/z* 3311, which corresponds to a nominal oligomer-to-charge ratio (n/z) of *1/1*, was notable due to its high intensity when compared to the surrounding oligomers. Analysis of the arrival time distributions (ATD) of this mass spectral peak (*Figure 1C*) revealed two features that correspond to higher-order CL oligomers. We conservatively assigned the longest arrival time feature (salmon) as an 8-mer with *z = +8*. This

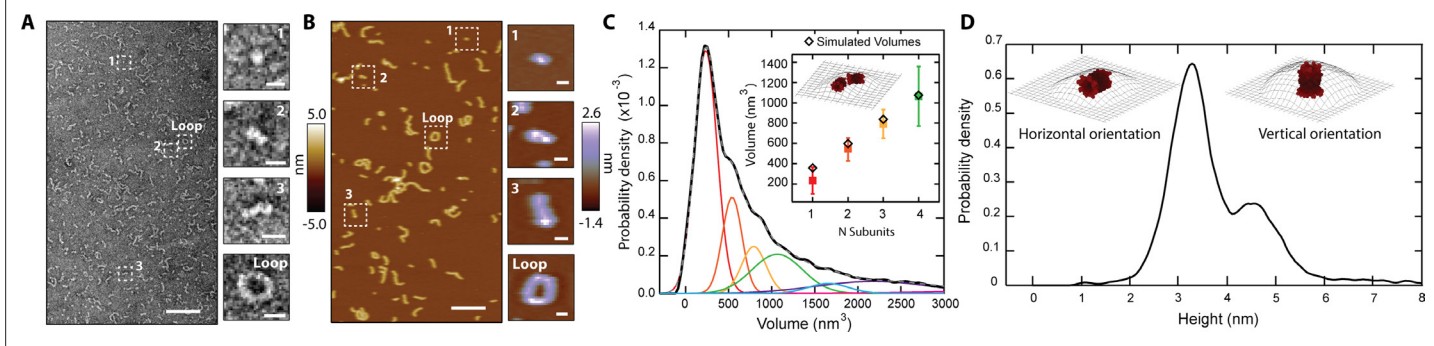

**Figure 2.** CL in solution forms progressively longer linear polymers and loops. (**A**) TEM reveals CL polymerization; scale bar = 100 nm. Features of growing complexity are magnified in the side panels; scale bar = 10 nm. (**B**) AFM imaging in fluid shows overall agreement with TEM data; scale bar = 100 nm. Four similar CL features are also highlighted in the side panels; scale bar = 10 nm. (**C**) Smoothed volume histogram of N=7838 individual features (solid black line) fitted with Gaussian distributions (solid colored lines, summary fit is the gray dashed line). The inset compares the experimental volume of the first four peaks (squares colored as in panel C) with simulated volumes (black diamonds) calculated from $N$ 8-mers. The cartoon shows an example of head-to-toe assembly of two 8-mers. Error bars for the peak positions represent the standard deviation. (**D**) When the height of the individual features was measured, it yielded a bimodal distribution, with heights of 3.2 nm and 4.6 nm. The insets show the two proposed orientations of the 8-mer. The topographic surface is overlaid to show convolution of the AFM tip.

The online version of this article includes the following figure supplement(s) for figure 2:

**Figure supplement 1.** Persistence length of CL polymers.

**Figure supplement 2.** Simulated monomer versus experimental subunit.

**Figure supplement 3.** Simulated CL polymer volumes are comparable to experimental volumes.

assignment was performed comparing the isotopic spacing and drift time of the features within the ATD with other mass spectral peaks (see *Figure 1—figure supplement 1*). Since a species with a high charge will travel faster than one with a lower charge, the fastest arrival time species (green) is proposed to be a large oligomer resulting from the self-assembly of the 8-mer. The ATD data therefore suggest that CL forms an 8-mer that self-assembles. Given that CL adopts helical structure in solution (*Moyes et al., 2016*), we built an atomistic model of the 8-mer by aligning CL's sequence into the eight-helix coiled coil formed by CC-Type2-II (PDB ID 6G67). After the resulting structure was energy minimized (*Figure 1—figure supplement 1*), the theoretical collisional cross-section of the model, 2500 Å$^2$, agreed with the experimental value of 2567 Å$^2$.

We used two additional biophysical methods to confirm and further investigate CL oligomerization in solution. We first performed analytical ultracentrifugation of CL (*Figure 1D*). The sedimentation velocity results reveal a low sedimentation peak, likely corresponding to a CL monomer, and several larger assemblies, in agreement with the IM-MS data analysis. We next used mass photometry (MP), a technique that allows mass determination of biomolecules with single-molecule resolution from their combined light scattering and reflection signature (*Asor and Kukura, 2022*). While MP is insensitive to the mass of an CL 8-mer (26.5 kDa), as the resolution of the instrument is limited to particles >50 kDa, MP has high sensitivity to larger species. *Figure 1E* displays MP data of CL, consisting of a main peak and a long tail. The main peak agrees with the mass of two bound 8-mers, while the larger mass peaks would correspond to the progressive assembly of 8-mers (see *Figure 1—figure supplement 2*). The long tail of the MP data reaches beyond 600 kDa, which would correspond to the assembly of tens of 8-mers. Taken together, the four techniques reveal that CL readily forms large oligomers in solution and suggest that the 8-mer is the seed for CL self-assembly into large structures.

We employed microscopy to resolve the assemblies that CL forms in solution. We first performed negative-stain transmission electron microscopy (TEM), and observed that CL does not form amorphous aggregates, but instead assembles into linear structures (*Figure 2A*). The TEM data revealed a basic structural unit, which seemed to grow in a step-wise fashion into polymers (*Figure 2A*, side panels). The longer polymers curved, and in some cases closed in on themselves, forming a loop with diameter ≥10 nm. We further studied CL polymerization by atomic force microscopy (AFM) since this technique provides high resolution images of single particles and can also be readily used to image

membrane pores (*Dufrêne et al., 2017*). AFM images of CL adsorbed onto mica in buffer (*Figure 2B*) agreed with the TEM results and confirmed that CL polymerizes and can form loops. The bending stiffness of CL polymers was calculated from AFM images through persistence length, $L_p$, analysis ($L_p$ = 9 nm±2 nm, N=100, *Figure 2—figure supplement 1*). The bending stiffness of linear CL polymers is significantly lower than other biological polymers, like actin fibers (*Gittes et al., 1993*), explaining the ability of CL polymers to close into loops.

Examination of AFM images containing thousands of particles yielded a volume distribution with a sharp peak and a long tail (*Figure 2C*), which resembles the distribution observed in the MP data (*Figure 1E*). Such agreement suggests that the AFM results are robust and are not caused by spurious interactions with the AFM tip. The AFM volume histogram was deconvolved by fitting Gaussian distributions, revealing a primary peak and several larger sub-populations of larger volumes (*Figure 2C*). The volume increased in approximately constant steps between the populations (see side panels of *Figure 2B*), confirming that a basic intermediate oligomer grew first by dimerization, followed by sequential addition to further increase linear polymer length.

We measured the height of the individual particles observed by AFM, and the resulting histogram (*Figure 2D*) yielded two peaks, with heights of 3.2±0.4 nm and 4.6±0.5 nm. We reasoned that the bimodal height distribution could indicate two orientations (vertical and horizontal) of the basic structural unit that serves as an intermediate oligomer (*Figure 2B* inset, labeled '1'), which is probably the CL 8-mer identified by IM-MS (*Figure 1B*, *Figure 2—figure supplement 2*). We tested this possibility by using the CL 8-mer model (*Figure 1—figure supplement 1*) to generate simulated AFM images through morphological dilation of the estimated tip geometry for the 8-mer, either on its horizontal or vertical orientations (*Figure 2D*, insets). We found good agreement between the simulation [*heights* of 3.3 nm and 4.8 nm] and the two experimental peaks shown in *Figure 2D*. Interestingly, the *volume* of the modeled 8-mer in the horizontal orientation additionally agrees within uncertainty with the value of the basic AFM structural unit (*Figure 2C* inset). Taken together, these results support the notion that CL assembles into an 8-mer in solution that acts as an intermediate step in the self-assembly process.

Polymer growth of CL 8-mers in the horizontal orientation (lying flat on a surface) can occur by addition of a second 8-mer in two fundamental modes, side-by-side or head-to-toe. When we modeled the latter arrangement for the assembly of two, three, and four head-to-toe 8-mers (*Figure 2C*, inset), we found agreement between the simulated and experimental AFM volumes (*Figure 2C*, inset; *Figure 2—figure supplement 3*). Taken together, the data indicate that a CL polymer is formed when a basic subunit, which our results suggest is an 8-mer, is aligned in the longest dimension, and grows by sequential head-to-toe addition of additional subunits.

## CL forms two classes of membrane pores

CL mediates *C. albicans* infection by damaging the integrity of the plasma membrane of human cells (*Moyes et al., 2016*). It has been proposed that CL forms membrane pores (*Moyes et al., 2016*), but such structures have never been observed. We performed AFM imaging on supported lipid bilayers made of DOPC (1,2-dioleoyl-sn-glycero-3-phosphocholine) to gain insights into how CL causes membrane disruption. We observed that in the presence of CL, bilayers indeed exhibited punctate depressions commonly associated with pores (*Figure 3A*). Though the AFM tip is sharp (nominal radius ~8 nm), the pores' radii were often smaller than the tip, preventing passage all the way through the 4-nm-thick membrane, which cause artificially shallow readings (*Pittman et al., 2018*; *Schaefer et al., 2022*; *Figure 3—figure supplement 1*).

The CL pores fell into two broad categories: simple depressions (*Figure 3A*), and complex pores surrounded by a rim of discrete protrusions of an approximate height of 0.3 nm above the bilayer surface (*Figure 3B*). To differentiate between the two types of pores, they are henceforth referred to as 'unrimmed' (*Figure 3A*) and 'rimmed' (*Figure 3B*) pores, respectively. Both types of pores were free to diffuse in the bilayer (*Figure 3—figure supplement 2*), indicating that CL is not immobilized by interactions with the mica surface. Occasionally rimmed and unrimmed pores were observed in the same image area (*Figure 3—figure supplement 3*), suggesting that variations in imaging conditions were not responsible for driving the system between rimmed and unrimmed states. Rimmed pores generally have a smaller area than unrimmed pores (*Figure 3C*). Unrimmed pores have a broad area distribution, with the main population at 41 nm$^2$ (*Figure 3C*, gray), while the rimmed pores have a

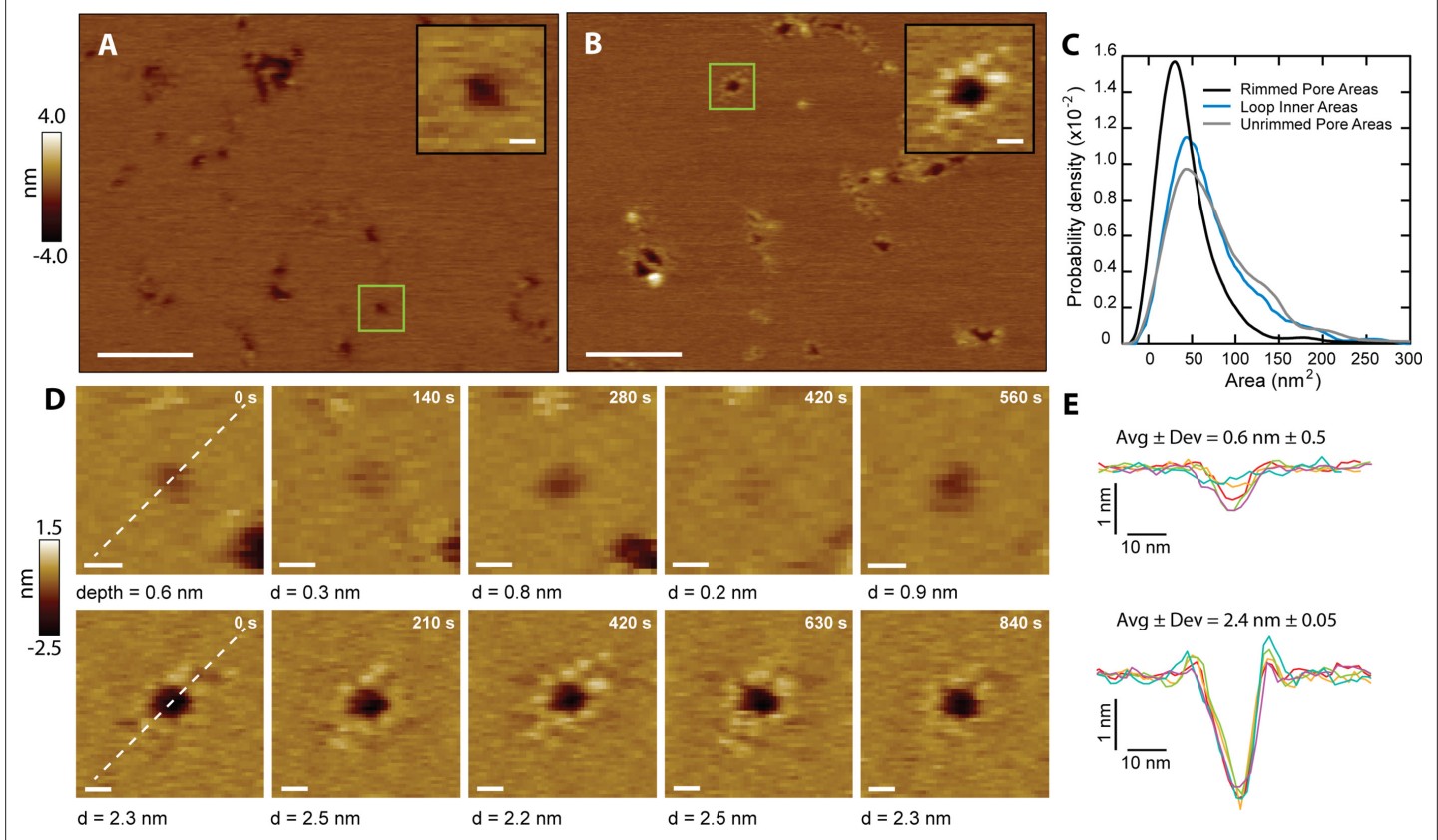

**Figure 3.** CL forms two types of membrane pores. Representative images showing predominantly unrimmed (**A**) and rimmed (**B**) pore features in supported DOPC membranes; scale bars = 100 nm. The insets show zoomed views of green boxed features; scale bars = 10 nm. (**C**) Area histograms show that unrimmed pores (gray line) display a broad peak at $41\pm25$ nm$^2$ (mean ± S.D.) with a shoulder at ~130 nm$^2$, similar to loops found in solution ($38\pm21$ nm$^2$, aqua blue line). In contrast, the rimmed pores (black line) exhibit a narrower distribution with a smaller area ($26\pm19$ nm$^2$) ($N_{unrimmed}$ pores = 1468, $N_{rimmed\ pores}$ = 492, $N_{loops}$ = 261). (**D**) Unrimmed (*top*) and rimmed (*bottom*) features were imaged over several minutes; scale bars = 10 nm. (**E**) Line scans, marked as white lines in the previous panel, show the dynamics of the pore profile in both cases. Different colors were used for each image. The average depth (Avg) and relative deviation (Dev), defined as the standard deviation of the depth divided by Avg, are listed.

The online version of this article includes the following figure supplement(s) for figure 3:

**Figure supplement 1.** Quantification of pore depth.

**Figure supplement 2.** CL pores exhibit lateral dynamics in the AFM supported membrane.

**Figure supplement 3.** Unrimmed and rimmed pores locally coexist in the membrane.

narrower distribution, and an area peak at 26 nm$^2$ (*Figure 3C*, black). Selected areas of the samples were imaged repeatedly to produce a time series of both unrimmed and rimmed pores. A representative unrimmed pore (*Figure 3D*, top row) shows dynamic behavior, with large variations over time. The pore depth visibly varies, flickering between deep and shallow states. On the other hand, the rimmed pore (*Figure 3D*, bottom row) appears deeper, and is more stable, as shown by the constant profile over time (*Figure 3E*). While the two types of pores are likely to cause membrane permeabilization, we propose that the presence of the rim endows the pore with increased stability, which could potentially be a mechanism to afford CL pores with enhanced membrane damaging capabilities.

## CL polymers insert into the membrane, and loops become membrane pores

We noticed that there were strong similarities between the shape and size of the loops that CL forms in the absence of membrane and the unrimmed membrane pores (*Figure 3C*, compare blue and gray lines). This agreement led us to hypothesize that CL loops could insert into membranes and become unrimmed pores. We could test this hypothesis as occasionally a patch of supported bilayer would

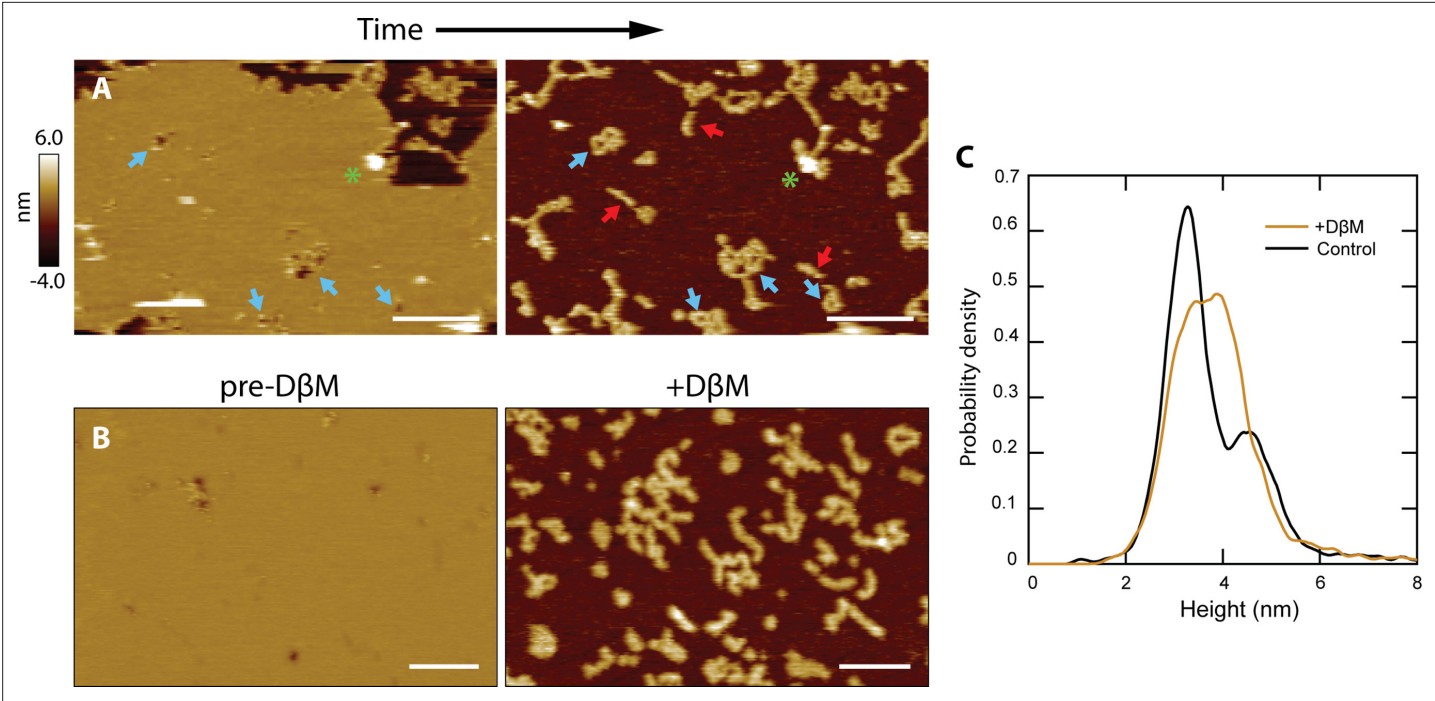

**Figure 4.** CL polymers and loops insert into membranes. (**A**) Lipid patches exhibiting pore-like features occasionally dissociated from the mica surface, revealing underlying structures similar to CL polymers in solution [see *Figure 2B*]; scale bars = 200 nm. Blue arrows indicate pore features and corresponding polymer loops/tangles. Linear features were revealed that were not previously observed when lipid was present (red arrows). A green asterisk draws attention to a tall positive feature that remains in both images, serving as a reference. (**B**) Detergent was used to forcibly remove lipid bilayers. The addition of DβM followed by rinsing revealed underlying CL structures. Scale bars = 100 nm. (**C**) A histogram of particle heights compares data of CL adsorbed in solution (black, N=7838) to the features remaining after addition of DβM to the CL +DOPC system (gold, N=8170). The control results were bimodal, indicating two overall 8-mer orientations. The features left after detergent removal of the membrane exhibited a broad peak roughly encapsulating both of the solution peaks.

The online version of this article includes the following figure supplement(s) for figure 4:

**Figure supplement 1.** Detergent effect on bare lipid and CL on a surface.

spontaneously dissociate from the underlying mica surface (*Figure 4A*). Upon imaging the exact same area again, we observed features similar to the CL polymers and loops detected in the absence of lipid bilayers. Indeed, loops, often inter-connected, were observed where pores were previously present (see blue arrows). Linear features similar to the CL polymers observed for solution samples were also present, which had been previously obscured by the presence of a bilayer. This observation indicates that linear polymers can insert into membranes without causing membrane disruption, but when they close into a loop they form an unrimmed membrane pore.

In order to replicate these spontaneous events in a deterministic manner, we used the mild detergent dodecyl beta-D-maltoside (DβM) to remove the lipid bilayer (*Milhiet et al., 2006*; *Rinia et al., 2001*). Once the detergent and solubilized lipid were rinsed away, the CL structures remaining were imaged (*Figure 4B*). Control experiments showed that DβM efficiently removed DOPC molecules from the mica support, but did not dissociate the CL polymers (*Figure 4—figure supplement 1*); however, our data suggest that detergent treatment might reduce the fraction of polymers that close into a loop (*Figure 4—figure supplement 1*). The histogram in *Figure 4C* reveals similarities between the height of CL features adsorbed to mica in solution, showing a bimodal population, and in membrane areas post-DβM treatment. In these latter samples, there was a broadening of the primary height peak, which can be attributed to binding between CL and the detergent. However, the increase in taller features could also be an indication that the vertical 8-mer orientation (*Figure 2D*) is favored in the presence of lipid membranes.

Our data indicate that loops can insert into membranes and form unrimmed pores. These might later mature into the rimmed pores, which have better defined dimensions. The question then arises regarding what orientation, head-to-toe or side-by-side, does the CL 8-mer adopt in the two types

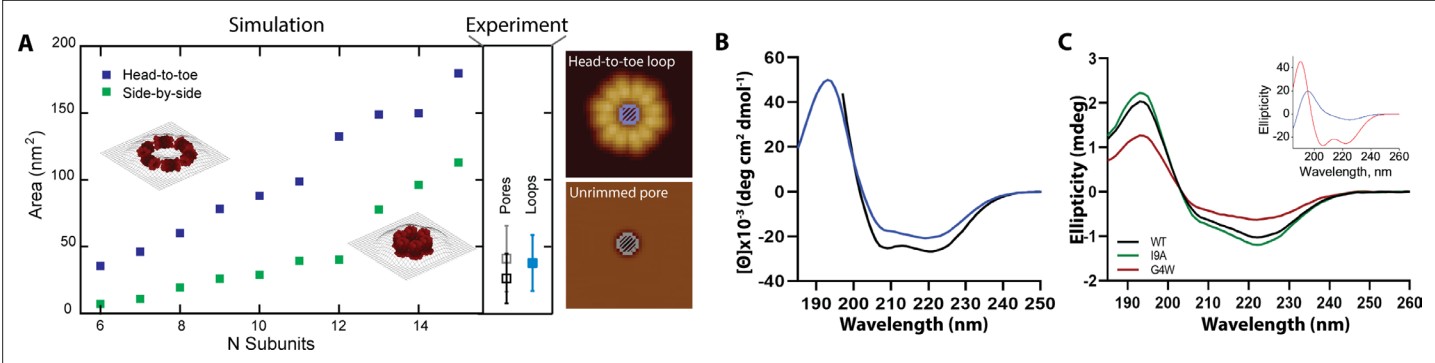

**Figure 5.** Rimmed pores appear upon a conformational rearrangement. (**A**) CL pores were simulated using two possible arrangements of the 8-mer subunit, side-by-side (green) and head-to-toe (dark blue). The particle area is plotted versus the number of subunits. The insets show simulations for N=6 in both orientations. Closed loops with less subunits were not geometrically feasible. The side panel shows the experimental areas for unrimmed pores (gray symbol), rimmed pores (black), and the loops found in solution (aqua blue); error bars represent standard deviation. A cartoon depicts which measurements are being compared, the inner loop area (top) and the pore area (bottom). (**B**) Standard circular dichroism (CD) of CL in buffer (blue) and in the presence of lipid vesicles (black). (**C**) OCD data in supported membranes for CL WT (black), I9A (green), and G4W (red). The inset shows theoretical OCD curves for a TM helix (blue) or an α-helix aligned parallel to the membrane plane (red).

The online version of this article includes the following figure supplement(s) for figure 5:

**Figure supplement 1.** Direct comparison of simulated loops with AFM data.

of pores. To investigate this question, we modeled loops of *N* subunits in the two 8-mer orientations, head-to-toe and side-by-side (***Figure 5A***). Comparison between the experimental inner areas and the models (***Figure 5A***, ***Figure 5—figure supplement 1***) suggest that the *loops* found in solution, similar to the *linear polymers*, assemble into a head-to-toe fashion. The average loop would be composed of six head-to-toe 8-mers, for a mass (159 kDa) that corresponds with the start of the tail in the MP distribution (***Figure 1E***).

We further studied CL morphology after membrane insertion by analyzing the features that remained after spontaneous lipid dissociation (***Figure 4A***). For unrimmed pores, the AFM data also corresponded to a head-to-toe 8-mer arrangement (***Figure 5A***). Loops with a side-by-side orientation were rare in the solution data, but were common in CL samples that had been exposed to lipid. Indeed, comparison between the model and the rimmed pores data suggest that in these pores the 8-mers switch from the head-to-toe orientation found in the unrimmed pores (gray square, side panel of ***Figure 5A***), to a side-by-side orientation (black square). Specifically, the modeled areas predict that unrimmed pores contain six to eight head-to-toe 8-mers, while rimmed pores are formed by six to twelve side-to-side 8-mers. Therefore, we propose that a rimmed pore (***Figure 3B***) appears when the 8-mer subunits in an unrimmed pore rotate 90 degrees, switching from the head-to-toe to the side-by-side arrangement (compare insets in ***Figure 5A***).

We interrogated the results of the modeling using circular dichroism (CD). ***Figure 5B*** shows that CL forms, as expected (***Moyes et al., 2016***), an α-helical structure in solution, as evidenced by the spectral minima at 208 and 222 nm. We also observed that the presence of lipid vesicles promotes modest additional helix formation, possibly by growth of the α-helix and its ends. The orientation of the CL α-helix with respect to the membrane plane can be determined using oriented CD (OCD), which is performed using supported lipid membranes (***Nguyen et al., 2019***). OCD can discriminate between helices aligned along the membrane plane, and transmembrane (TM) helices (***Wu et al., 1990***; ***Figure 5C***, inset). The OCD data of CL displayed low intensity, and a minimum at ~225 nm and a maximum at ~195 nm, similarly to the expected result for TM helices (***Alves et al., 2018***; ***Stefanski et al., 2021***). The OCD spectrum therefore indicates that the predominant orientation of helices in CL in membranes was inserted across the membrane, in a TM orientation. This is the proposed helical alignment that is expected in the side-by-side 8-mer arrangement of the rimmed pores. However, the presence of a depression at ~210 nm agrees with the presence of a population of α-helices oriented along the plane of the membrane, like those expected for the unrimmed pore. Therefore, the OCD result supports the presence of a combination of rimmed and unrimmed CL pores with the respective helical orientations predicted by the AFM modeling.

## Mutational analysis of CL provides mechanistic insights

Our experimental data and modeling indicate that CL 8-mers in solution polymerize in a head-to-toe fashion. We sought to further validate this hypothesis by testing mutations in the CL sequence. A head-to-toe assembly of a parallel helical bundle implies interaction between the N-terminal ($N_t$) helical ends of an 8-mer with the C-termini of the adjacent subunit (*Figure 6—figure supplement 1*). Our 8-mer model predicts that the $N_t$ is slightly kinked at residue G4 (*Figure 6A*). Since glycine residues confer flexibility to α-helices (*Choy et al., 2003*), we reasoned that the mutation of residue G4 (*Figure 6A*) might affect CL self-assembly. We replaced this residue with a bulky tryptophan side chain to maximize the expected disturbance. We first tested the ability of the resulting G4W CL variant to polymerize in solution using MP, and observed that G4W formed few large assemblies (*Figure 6B*). The residue G4 lies at the outer surface of the hollow cylinder that the 8-mer forms. We sought to test the effect of a residue at the opposite side of the helix, which is predicted to form the core of the structure. We therefore tested the I9A mutation. The MP results show that the I9A variant, on the other hand, self-assembles efficiently in solution (*Figure 6B*).

We next tested the two variants using AFM to visualize the morphology of the polymers they formed. The AFM results agreed with the MP, as I9A formed long polymers and closed loops, while G4W formed few large structures (*Figure 6C*, compare top and bottom panels). We quantified the AFM images by measuring the volume of all particles, and compared these results with AFM data of WT CL. The resulting *Figure 6D* shows that G4W does not efficiently assemble into large (>1000 nm$^3$) structures, while WT and I9A do. However, only the I9A variant forms a significant number of very large assemblies (>3000 nm$^3$), which generally corresponded with over-sized loops (*Figure 6C*, side panel). Since the AFM data (*Figure 4*) indicate that linear polymers need to close to form membrane pores, a key parameter for membrane disruption would be the ability of polymers to close into loops. We determined the frequency of loop formation for the three peptides (*Figure 6E*), and observed that there is overall agreement between the ability to form polymers and the loop formation propensity. Consistent with the model that loops in solution become pores, we observed that I9A formed larger membrane pores than WT (*Figure 6—figure supplement 2*), while we did not observe membrane pores formed by G4W (*Figure 6F*). Additionally, the higher I9A pore area has a corresponding increase in the inner area of the loops compared to WT CL (*Figure 6—figure supplement 2*), supporting the hypothesis that the pore architecture is defined by the loops.

We studied next the ability of CL WT and the two variants to disrupt the integrity of POPC (1-palmitoyl-2-oleoyl-sn-glycero-3-phosphocholine) vesicles in suspension. Specifically, we carried out an assay where membrane damage is measured by the fluorescence de-quenching that occurs when the dye calcein is released from vesicles. *Figure 6G* shows the kinetics of membrane disruption caused by the addition of WT and the two CL variants. We observed that G4W did not induce membrane disruption, in agreement with the lack of membrane pores observed by AFM. WT CL efficiently disrupted membranes, but I9A was the most efficacious at disrupting membrane integrity, as expected from the high loop formation probability (*Figure 6E*) and the presence of larger membrane pores (*Figure 6—figure supplement 2*). The differences in leakage were consistent over a variety of peptide concentrations in the nanomolar range (*Figure 6G*). To study why the G4W mutation impacted the ability to disrupt membranes, we performed CD and OCD experiments on the variants. CD revealed that G4W showed reduced helical structure in solution and in the presence of POPC liposomes (*Figure 6—figure supplement 3*). *Figure 5C* also shows changes in the OCD spectrum for G4W in comparison to WT and I9A, which indicate substantial differences in how this variant interacts with lipid membranes. When we performed SDS-PAGE of the variants, we observed that I9A displayed higher levels of oligomeric intermediates (*Figure 6H*). This result suggests that the higher tendency of I9A to polymerize is due to its enhanced oligomerization ability. G4W, on the other hand, displayed similar levels of oligomers as WT. We therefore speculate that a W present at the $N_t$ might impair the ability of two oligomers to bind. This effect would therefore reduce the formation of polymers, and explain the loss-of-function effect of this mutation. Overall, the mutational data confirm that CL uses a novel molecular mechanism, where the peptide first self-assembles in solution to form linear polymers that close into pore-competent soluble loops. The results of the loss-of-function G4W variant revealed that formation of polymers in solution is a necessary step for the formation of membrane pores.

CL pores damage the plasma membrane of epithelial cells. This attack triggers the activation of danger response signaling, which induces phosphorylation of MAPK phosphatase 1 (MKP1) and

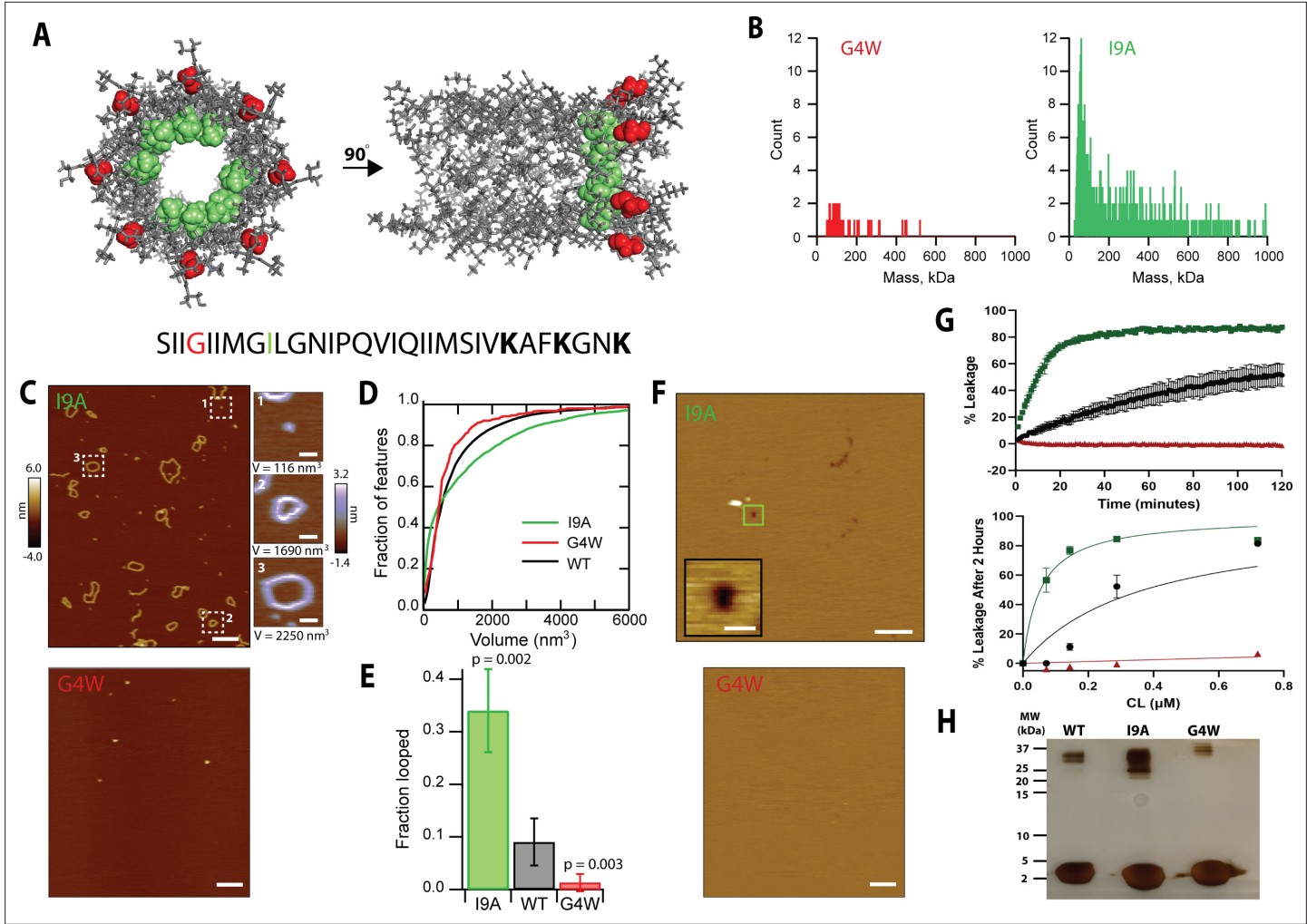

SIIGIIMGILGNIPQVIQIIMSIVKAFKGNK

**Figure 6.** Mutational analysis indicates that polymerization in solution is required for pore formation. (**A**) The position of the residues G4 (red) and I9 (green) are shown for two different orientations of the 8-mer model. The CL sequence highlighting the position of the mutations is shown at the bottom. (**B**) Mass photometry shows that the G4W variant has a lower tendency to form large assemblies than I9A in solution. (**C**) AFM data of CL variants in the absence of membranes. The I9A variant exhibits increased loop features, while G4W does not form polymers; scale bar = 100 nm. Representative I9A features are selected from the image: a protomer, a loop, and a large loop; scale bars = 10 nm. (**D**) The volumes of the features observed by AFM were calculated, and are shown as accumulated fraction. Data are shown for CL WT (N=7838 features), I9A (N=3609), and G4W (N=190). (**E**) The fraction of all polymers that close into a loop was calculated for all variants. Error bars are standard deviations (n=3 independent experiments); p-values are the result of a Student's t-test comparing the I9A and G4W data to the WT. (**F**) I9A CL forms membrane pores, while pores were not observed for G4W; scale bars = 100 nm for full images, 10 nm for inset. (**G**) Liposome dye release assay shows that CL variants display different membrane disruption. *Top* panel shows a time course of dye release for CL WT (black), I9A (green), and G4W (red) at 333 nM, and the *bottom* panel shows the percentage of dye release after 2 hrs for different peptide concentrations. N=3–4 and error bars are S.D. (**H**) SDS-PAGE of variants. The 16% tricine gel was silver-stained. Samples were incubated at 37 °C. The position of molecular weight markers is shown. The lower mobility of the G4W oligomer results from the corresponding differences in molecular weight ($MW_{CL-WT}$=3310 Da, and $MW_{CL-G4W}$ = 3439 Da).

The online version of this article includes the following source data and figure supplement(s) for figure 6:

**Source data 1.** Raw mass photometry data corresponding to *Figure 6B*.

**Source data 2.** Uncropped gel corresponding to *Figure 6H*.

**Figure supplement 1.** Head-to-toe model of CL 16-mer (dimer of octamer).

**Figure supplement 2.** Particle volume and pore area histograms of CL variants.

**Figure supplement 3.** CD spectra of WT and variant CL in the presence and absence of lipid vesicles.

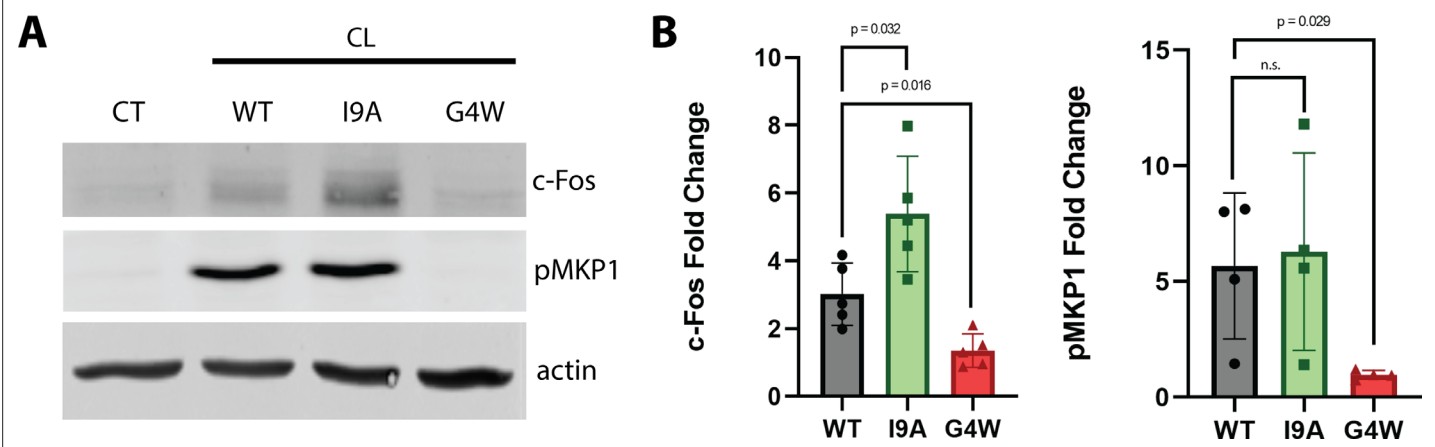

**Figure 7.** The polymerization-defective G4W variant does not cause danger response signaling in epithelial cells. (**A**) Representative western blots of oral epithelial cells (TR146) probed for c-Fos expression and phosphorylation of MKP1 after 2 hr treatment with WT and variant CL. (**B**) Quantification of c-Fos and MKP1 phosphorylation was used to assess danger response signaling. Data were normalized to values obtained in control conditions. Actin was used as a loading control. N=4, and bars are the S.D.

The online version of this article includes the following source data for figure 7:

**Source data 1.** Uncropped western blot of TR146 cell lysates treated with WT and variant CL for two hours probed with antibodies for (A) c-Fos, actin and (B) pMKP1.

subsequent overexpression of the transcription factor c-Fos (*Richardson et al., 2018b*; *Richardson et al., 2018a*). When we treated the oral epithelial cell line TR146 with WT CL, we observed robust increases in c-Fos (threefold) levels and phosphorylation of MKP1 (5-fold) (*Figure 7*), as expected (*Moyes et al., 2016*). The I9A variant also induced strong MKP1 phosphorylation, but significantly higher c-Fos expression. Conversely, G4W CL failed to activate the danger response signaling pathway altogether, revealing an overall agreement between the effect of CL mutations in biophysical and cellular assays. Our data therefore indicate that mutations that alter polymerization impact epithelial cell damage response signaling. Taken together, these results suggest that polymerization of CL in solution is a major factor defining *C. albicans* infection.

## Discussion

Pore-forming toxins (PFTs) populate different states on their way to form membrane pores. PFTs are typically released as soluble monomers that later bind to the target lipid bilayer. *After* membrane binding, PFT monomers find each other and oligomerize (*Leung et al., 2014*; *Verma et al., 2021*), often forming membrane-anchored pre-pores that later transition into the mature pores that permeabilize membranes (*Mondal and Chattopadhyay, 2020*; *De Schutter et al., 2021*). Here, we show that CL follows a different molecular mechanism for pore formation. Multiple lines of evidence indicate that CL forms both linear and closed loop polymers *before* binding to the membrane (*Figure 8*). Further, acting as soluble pre-pores, closed loop CL polymers insert into lipid bilayers to cause membrane damage without any observable conformational change.

It is interesting to compare the topography and mode of action of CL pores to those formed by other PFTs and other proteins. AFM imaging revealed that CL (3.3 kDa) assembles into intermediate size membrane pores, of diameter ≤80 Å, deduced from area distribution peaks <50 nm$^2$ (*Figures 5A and 3C*) and assuming circularity, which matched many of the pores observed. This diameter is smaller than the well-studied aqueous pores formed by the proteins gasdermin-D (*Mulvihill et al., 2018*), suilysin (*Leung et al., 2014*) and perfringolysin O (*Ramachandran et al., 2002*; *Czajkowsky et al., 2004*), of molecular weights in the 50–60 kDa range, but is probably larger than pores formed by α-hemolysin (diameter: 14–46 Å, 33 kDa) (*Song et al., 1996*). Further differences exist between the assembly mechanism of these proteins and the mode of action of CL. Multiple PFT families such as hemolysins form large pre-pore structures at the membrane surface prior to insertion and pore maturation (*Dal Peraro and van der Goot, 2016*). In contrast, CL unrimmed pores present no features

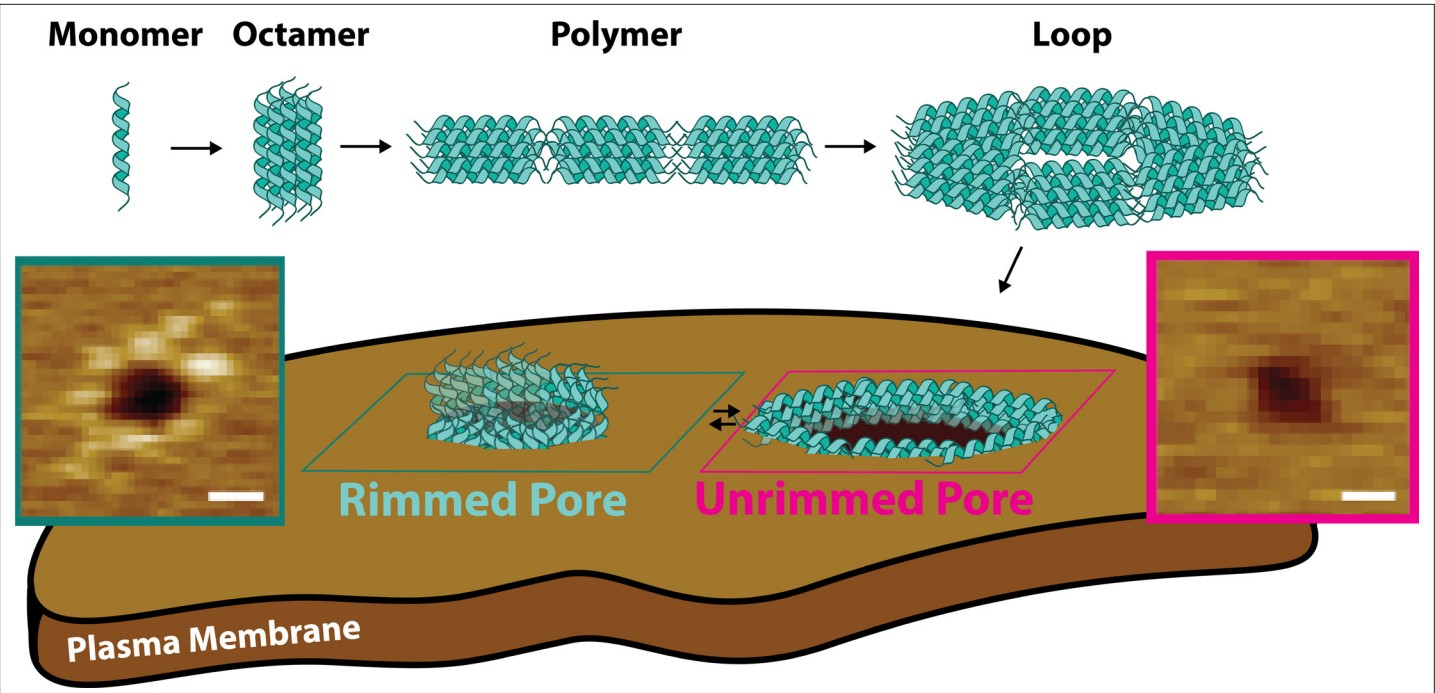

**Figure 8.** Candidalysin utilizes a novel mechanism of pore formation to attack human cell membranes. The CL monomer oligomerizes into an intermediate oligomer, which we propose is 8-mer. CL polymers are formed when the oligomer assembles in a head-to-toe fashion. Polymers can close into loops, which have the ability to insert into membranes forming unrimmed pores. Upon insertion, pores exist in a dynamic equilibrium between unrimmed and rimmed pores, which cause cellular toxicity. Scale bars in the AFM images are 10 nm.

above the upper leaflet of the bilayer. Instead, these dynamic structures appear topographically similar to those induced by the antimicrobial peptide melittin and its derivatives (*Pittman et al., 2018*; *Hammond et al., 2021*; *Kim et al., 2019*). The rimmed CL pores are quite distinct from the unrimmed pores. Not only are they more stable on the timescale of AFM imaging (>0.1 s), but rimmed pores have a punctate pattern about their periphery. Such structure is reminiscent of the pores formed by the N-terminal segment of gasdermin-D (*Mulvihill et al., 2018*) however, the overall size of the CL rim is smaller, only reaching ~0.3 nm outside each side of the bilayer surface. This height corresponds to the difference between the 8-mer (4.6 nm) and the DOPC bilayer thickness (4.0 nm, as measured by AFM, *Figure 3—figure supplement 1*). This observation suggests that the pore rims arise when the ends of the TM helices of the 8-mer stick out of the bilayer surface upon 90 degree rotation of the 8-mer in the unrimmed pore, which transitions from the head-to-toe into the side-to-side arrangement (*Figure 8*). Some rimmed pores were decorated with protrusions covering the whole pore periphery, forming a corona (*Figure 3B*). However, other pores were only partially covered with puncta (*Figure 4A*). This observation suggests that maturation into rimmed pores occurs progressively and is not an all-or-none process.

Pore maturation for perfringolysin O (*Czajkowsky et al., 2004*; *Nelson et al., 2008*) and suilysin (*Leung et al., 2014*), involves a *vertical collapse* that allows formation of a TM pore. In contrast, the rearrangement of the 8-mer orientation that leads to rim formation causes CL to protrude from the bilayer and could be defined as a *vertical swell* between the two types of pores. The reason why pore maturation in CL occurs is not clear, but it might be related to the three K residues located at the $C_t$ (*Figure 6A*). In the unrimmed pores, the head-to-toe orientation of the 8-mers would likely place the positive charges of the K side chain in contact with lipid molecules, resulting in a meta-stable configuration. However, in the vertical 8-mer orientation found in the rimmed pore, the K residues are expected to be solvent-exposed, resulting in a more stable conformation. An additional difference with some pore-forming proteins is that imaging of bilayers in the presence of CL did not show open polymers. However, bilayer removal revealed that linear CL polymers insert into the membranes, but they cannot be readily observed as they do not disturb the bilayer surface or protrude significantly from it. This behavior contrasts with the arcs formed by suilysin, as these rigid open polymers were

able to remove lipid molecules from the bilayer to cause perforations (*Leung et al., 2014*). Finally, we compare the CL mechanism with the membrane attack (MAC) system that forms membrane pores in pathogens. MAC employs the 71 kDa C9 protein, which polymerizes in solution prior to forming pores (*Song et al., 1996*; *Dudkina et al., 2016*; *DiScipio and Hugli, 1985*). However, C9 needs other components of the MAC system inserted into the membrane (e.g. C7 and C8), which then serve to recruit multiple C9s. Hence significant distinctions remain between the pore-forming mode of action of CL and that of the MAC system.

## Conclusions

Here, we elucidate how CL forms the membrane pores that damage cells infected by *C. albicans*. Our data show that CL preassembles into a pore-competent loop conformation in solution. To the best of our knowledge, this process represents a novel mechanism of pore formation, since other PFT that form aqueous pores require binding to the plasma membrane as monomers to self-assemble. The assembly of CL into an insertion-competent state in solution might provide an infectivity advantage for *C. albicans*, as it could facilitate faster membrane damage. At the same time, we propose that this assembly mechanism might constitute a therapeutic opportunity to inhibit CL pore formation. Targeting membrane pores presents several drug delivery drawbacks, and it is therefore significantly more challenging than a soluble target. However, drug molecules that prevent CL polymerization in solution could be used as therapeutics to fight *C. albicans* infection. Specifically, a molecule that inhibits the formation of CL loops would be expected to prevent the assembly of the membrane pores that damage human epithelial cells.

## Limitations of the study

Our data suggest that the oligomeric state of the basic assembly unit is a CL 8-mer, as indicated by the native mass spectrometry data and this assignment is consistent with AFM volume analysis. However, it is intrinsically challenging to precisely determine the stoichiometry of large oligomeric assemblies. We therefore cannot rule out that CL assembles instead through an intermediate of different stoichiometry (e.g., a 7-mer or a 9-mer). High resolution methods will also be required to confirm the directionality that the CL oligomers adopt in the polymer, to confirm that they are aligned head-to-tail. We additionally identified the loss-of-function variant G4W, and the possible gain-of-function I9A. Future studies that unravel how these mutations affect the biophysical parameters that govern peptide self-assembly and membrane interaction, are expected to shed light on the specific reasons behind the disparate effects of the mutations. Additionally, these investigations could allow us to determine if CL pore formation is indeed under kinetic control, as we suspect.

## Materials and methods

### Key resources table

| Reagent type (species) or resource | Designation | Source or reference | Identifiers | Additional information |
|---|---|---|---|---|
| Software, algorithm | Agilent 6560 IMS-QTOF Software | Agilent Technologies | | |
| Software, algorithm | Origin Pro | Origin | | Used to graph IMS-MS data |
| Software, algorithm | Visual Molecular Dynamics (VMD) | http://www.ks.uiuc.edu/Research/vmd/ | | |
| Software, algorithm | EMBOSS | EMBL-EBI | | |
| Software, algorithm | GROMACS 4.6.7 | GROMACS | | |
| Chemical compound, drug | Ammonium Acetate, Ultrapure | VWR | 631-61-8 | |
| Chemical compound, drug | LC-MS Water | Sigma-Aldrich | 7732-18-5 | |

*Continued on next page*

*Continued*

| Reagent type (species) or resource | Designation | Source or reference | Identifiers | Additional information |
|---|---|---|---|---|
| Cell line (*Homo sapiens*) | Buccal squamous cell carcinoma | ECACC | TR146 | ECACC 10032305 |
| Antibody | Goat-anti mouse IRDye 680 (Goat polyclonal) | LI-COR | 925–68070 | 1:10,000 |
| Antibody | Phospho-MKP1 (Rabbit monoclonal) | Cell Signaling Technology | 2857 S | 1:1,000 |
| Antibody | c-Fos (Rabbit monoclonal) | Cell Signaling Technology | 2250 S | 1:500 |
| Antibody | Goat-anti rabbit IRDye 800 (Goat polyclonal) | LI-COR | 926–32211 | 1:10,000 |
| Antibody | β-actin (mouse monoclonal) | ABCAM | ab6276 | 1:1000 |
| Peptide, recombinant protein | Wild-Type Candidalysin | Peptide 2.0 | CL | |
| Peptide, recombinant protein | I9A Candidalysin | Peptide 2.0 | I9A | |
| Peptide, recombinant protein | G4W Candidalysin | Peptide 2.0 | G4W | |
| Chemical compound, drug | POPC | Avanti Polar Lipids | 850457 C | |
| Chemical compound, drug | DOPC | Avanti Polar Lipids | 850375 P | |
| Chemical compound, drug | Calcein | MP | 02190167-CF | |
| Software, algorithm | Discover MP | Refeyn version 2.2.0 | | |
| Software, algorithm | Acquire MP | Refeyn version 2.2.0 | | |
| Software, algorithm | SEDFIT | NIH | | |
| Software, algorithm | Image Studio | LI-COR | | |
| Chemical compound, drug | n-Dodecyl-beta-D-Maltoside | Millipore Sigma | D4641 | DBM |
| Software, algorithm | Igor Pro 7.08 | Wavemetrics | | |
| Software, algorithm | Igor Pro 6.38 | Wavemetrics | | |
| Software, algorithm | Asylum AFM Software | Oxford instruments | | Version 16 |
| Other | AFM tips | Olympus | | See AFM Materials and methods Section |

## Peptide preparation

Wild-type (sequence: SIIGIIMGILGNIPQVIQIIMSIVKAFKGNK) and variant candidalysin were synthesized employing solid-phase synthesis by Peptide 2.0 (Chantilly, VA) and HPLC-purified. Purity (>95%) was checked by MALDI-TOF and analytical HPLC. Lyophilized stocks were resuspended in MilliQ $H_2O$, stored at –80 °C, and diluted in the desired buffer at the time of experimentation.

## Ion-mobility mass spectrometry

Peptide samples were directly injected via a syringe pump into a JetStream ESI nebulizer. Mass spectral data and ion mobility measurements were collected using an Agilent 6560 IMS-QTOF mass spectrometer. The peptides were ionized in positive-mode and ions were subsequently pulsed into a helium-filled drift cell in a multi-field fashion (ΔV=890, 790, 690, 590, and 490 V). The pressure of the drift cell was maintained at 3.940 Torr with the pressure differences between the drift cell and trap funnel being approximately 300 mTorr. An exit funnel and hexapole ion guide focused the ions into the QTOF mass spectrometer. Mobility data were collected over the course of 5.2 min (additional instrument parameters are given in *Supplementary file 1*). Arrival time distributions were extracted with the Agilent IM-MS Browser software and graphed with Origin Pro.

The ion's arrival time can be related to the change in drift cell voltage (ΔV) to determine the ion's reduced mobility ($K_0$). To calculate the momentum transfer collision integral, which approximates the experimental collisional cross section ($\sigma_{exp}$), the Mason-Schamp equation was used (*Mason and McDaniel, 1988*):

$$\sigma_{exp} \approx \frac{3ze}{16N_0} \frac{1}{K_0} \sqrt{\frac{2\pi}{\mu k_B T}} \qquad (1)$$

where z is the ionic charge, e is the elementary charge, $N_0$ is the number gas density, μ is the reduced mass of the ion-buffer gas pair, $k_B$ is the Boltzman constant, and T is the temperature of the buffer gas. Theoretical collisional cross sections were calculated using the trajectory method available in the Mobcal package (*Mesleh et al., 1996*).

## Molecular dynamics simulation

An atomistic model of CL 8-mer was built by aligning the peptide sequence to that of the eight-helix coiled coil CC-Type2-II (PDB ID 6G67) (*Rhys et al., 2018*) using EMBOSS (*Madeira et al., 2019*). The initial structure was minimized using GROMACS 4.6.7 (39) during 25 ns and the CHARMM27 force field. Molecular dynamics (MD) simulation of the two interacting 8-mers was performed using the same package and force field. The initial structure was minimized using the steepest decent method and solvated in a TIP3P cubic water box (a=14.03 nm). Chloride anions were added to neutralize the charges. Solvent and volume equilibration simulations in NPT ensemble (T=300 K and *P*=1 bar) were performed to optimize the box size, followed by 45-ns NPT simulations at 300 K. The LINCS algorithm (*Hess, 2008*) was employed to constrain bonds between heavy atoms and hydrogen, and the SETTLE algorithm (*Miyamoto and Kollman, 1992*) was used for water molecules. These constraints allowed an integration time step of 2.0 fs. The electrostatic and dispersion forces were computed with a real space cutoff of 1.2 nm, and particle mesh Ewald method (*Darden et al., 1993*) was used to treat long-range electrostatics. The temperature was maintained by the Nose-Hoover thermostat. The temperature and pressure coupling constants were 0.1 and 1.0 ps, respectively. The equations of motion were integrated according to the leap-frog algorithm.

## Analytical ultracentrifugation

Sedimentation velocity experiments were performed at 20 °C with an An-60Ti rotor in a Beckman Optima XL-I ultracentrifuge at 35,000 RPM. Sedimentation was followed by absorbance at 230 nm and the continuous sedimentation coefficient distribution was obtained using SEDFIT. The following parameters were used for analysis: frictional ratio of 1.2, partial specific volume of 7.3 ml/g, buffer density of 0.9982 g/ml, and buffer viscosity of 1.002 mPa s.

## Mass photometry

WT and variant CL were diluted to 333 nM in the AFM imaging buffer (10 mM HEPES, 150 mM NaCl, pH 7.3) prior to measurement. Experiments were performed on clean glass coverslips using a Mass Photometer (OneMP, Refeyn). Videos were recorded for 120 s and analyzed on DiscoverMP (Refeyn,

version 2.2.0) to determine the molecular mass. The molecular mass was obtained by comparison with known protein standards measured on the same day.

## TEM

Transmission electron microscopy was performed using 2 µM CL samples in AFM imaging buffer incubated on carbon grids for 2 minutes at 25 °C, negatively stained with uranyl acetate, and imaged at 120kV (JEOL, JEM-1400).

## Liposome preparation

Lipids were purchased from Avanti Polar Lipids, Alabaster, AL. POPC (1-palmitoyl-2-oleoyl-glycero-3 -phosphocholine), DOPC (1,2-dioleoyl-sn-glycero-3-phosphocholine) stocks were prepared in chloroform and stored at –20 °C. Lipids were dried under argon gas and stored in a vacuum overnight prior to use in experiments. Lipid films were resuspended as described below. Large unilamellar vesicles (LUVs) were prepared using a Mini-Extruder (Avanti Polar Lipids, Alabaster, AL) with a 100 nm filter (Whatman, United Kingdom) (*Nguyen et al., 2015*; *Scott et al., 2015*).

## Atomic force microscopy

A sample of one hundred milligrams of DOPC in chloroform was divided into microcentrifuge tubes and dried under argon gas to form uniform films. The tubes were incubated overnight in a vacuum chamber. A dry mechanical roughing pump (XDS5, Edwards) was used to minimize the probability of contamination (*Schaefer et al., 2022*). Samples were back-filled with argon, sealed, and stored at –20 °C. At the time of extrusion, AFM imaging buffer was added to swell the lipids. The lipid solution was extruded (Liposofast, Avestin) through 100 nm membranes 25 times to form unilamellar vesicles. The solution was aliquoted and stored at –80 °C until the time of the experiment. A phosphorus assay was conducted to determine lipid concentration (*Stewart, 1980*). For AFM imaging of CL in solution, stock aliquots of CL were diluted in imaging buffer to the desired concentration (typically 333 nM). A 90 µL droplet was added to freshly cleaved mica and incubated for 10 min at 25 °C. Samples were rinsed from above the surface using buffer exchange (~100 µL of imaging buffer were exchanged over the surface 5–6 times). Images were collected in imaging buffer with biolever mini tips (Olympus, $k$~0.1 N/m, $f_o$~30 kHz in fluid) using tapping mode (Cypher, Asylum Research). Care was taken to keep the magnitude of the tip sample force to ≤100 pN during imaging deduced by comparing the free space tapping amplitude (~5 nm) to the imaging set point amplitude (~4 nm). Under such conditions, minimal protein distortion is expected (*Sanganna Gari et al., 2013*). As is typical in AFM, lateral image scales are significantly larger than the false color vertical scales.

To image pores, a method for forming lipid bilayers via vesicle rupture was adapted (*Schaefer et al., 2022*). An aliquot of DOPC liposomes was diluted to 300 µM and CL was diluted to specific concentrations The DOPC + CL solutions were incubated in a microcentrifuge tube for 10 min, with an additional mixing at the 5-min mark to encourage peptide diffusion (a pipette was inserted into the solution and the plunger depressed and released 5–6 times). Immediately after incubation in solution, 75 µL of the solution was deposited onto freshly cleaved mica and incubated for another 30 min. Material remaining in solution and loosely bound particles were removed by washing via buffer exchange (~100 µL of imaging buffer exchanged 5–6 times). All incubations were performed at room temperature (25 °C). Imaging was done in imaging buffer at ~35 °C. The underlying structure of CL peptides was revealed by removing the lipid bilayer with the non-ionic detergent dodecyl beta-D-maltoside (DβM). Stock DβM (196 mM) was diluted in the imaging buffer to 1.5 times the critical micelle concentration (CMC). After imaging pores, the AFM tip was lifted and 75 µL DβM added to the surface and incubated 15 min. To ensure all lipid and detergent molecules were removed from the surface, the samples were heavily rinsed (~100 µL of imaging buffer were exchanged 10–12 times). The remaining CL was then imaged in the imaging buffer. Analysis of solution data was performed using commercial software (Asylum Research, Inc). Pores were analyzed using the Hessian blob algorithm (*Marsh et al., 2018*). Probability density plots were generated using Epanechnikov kernels and the vertical axes were normalized (integrated to unity area). The number of Gaussian distributions was selected by minimizing the Bayesian information criterion for each model.

For AFM simulations, the tip was modeled as two overlapping spheres of different radii ($R$=8 and 4 nm) (*Schaefer et al., 2022*) and the CL 8-mer was positioned in specified orientations. For a single octamer, the geometry of the simulated image (volume = 360 nm$^3$) roughly corresponds with the geometry of the smallest subunits observed in AFM (volume of primary experimental peak ± σ=234 ± 130 nm$^3$). Polymeric arrangements of the octamer subunit were made by aligning them head-to-toe in linear or in loop conformations. The inset in *Figure 2C* compares the first four peak locations in the experimental data to the volumes of simulated curved arrangements of 1–4 subunits. Each of the simulated volumes agree with the experimental volume peaks within error. Curved simulations were chosen for this comparison because a linear model underestimates the measured volume due to a lack of curvature-dependent convolution in the simulation. To avoid volume degeneracies, only small (1–4 subunit) features are analyzed. The persistence length was calculated using Easyworm software (*Lamour et al., 2014*).

## Circular dichroism spectroscopy

The circular dichroism (CD) spectrum of candidalysin was collected in the presence and absence of POPC liposomes. LUVs were prepared by extruding resuspended lipid at 1 mM in sodium phosphate buffer pH 7.4. The final lipid concentration was 130 μM, and the final peptide concentration was 5 μM. CD spectra were measured in a Jasco J-850 spectropolarimeter with a 2 mm path length. A lipid blank and a buffer blank were subtracted from the lipid and lipid-free samples, respectively. Mean residue ellipticity (MRE) was calculated as:

$$[\theta]_{MRE} = \frac{\theta}{l \times (N-1) \times c \times 1000} \tag{2}$$

where $\theta$ is ellipticity (millidegrees), $l$ is cuvette pathlength (mm), $N$ is the number of amino acids in the peptide, and $c$ is the concentration of the peptide (M).

## Oriented circular dichroism

Two circular quartz slides (Hellma Analytics, Müllheim, Germany) were cleaned by submerging in piranha solution (75% $H_2SO_4$, 25% $H_2O_2$) for five minutes. POPC films were resuspended in 2,2,2-trifluoroethanol with or without CL, and this mixture was air-dried onto the slides for at least twelve hours at room temperature. The amount of total lipid on each slide was 85.3 nmol, and the amount of peptide on each slide 1.71 nmol, for a lipid to peptide ratio of 50:1. The deposited lipid was then rehydrated with PBS in a chamber containing saturated potassium sulfate, which produces a relative humidity of 96% to prevent the films from dehydrating. After at least 12 hr, the slides were loaded onto an OCD cell containing saturated potassium sulfate. OCD spectra were measured eight times on a Jasco J-815 spectropolarimeter. The OCD cell was rotated 45° between measurements to correct for any imperfections in the lipid bilayers. To obtain the final spectra, the eight measurements were averaged, and peptide-free lipid blanks were subtracted.

## Fluorescent dye-release assay

Dried lipid films were rehydrated with 50 mM calcein solubilized in 50 mM EDTA and 50 mM NaP$_i$ (pH 8) solution. LUVs were formed as described above, separated from free dye by gel filtration using a PD-10 column (GE Life Sciences, Chicago, IL), and diluted to a working lipid concentration of 144 μM. Peptide was added to the calcein entrapped LUVs to achieve the desired lipid:peptide molar ratios. The calcein fluorescence increase caused by dequenching was used as a proxy for CL-induced membrane leakage. A positive control was measured after each treatment, consisting of 0.016% w/v Triton X-100 (TX). A negative control of PBS *in lieu* of peptide treatment was used to calculate percent leakage:

$$\text{Normalized Leakage} = \frac{(F_{CL} + F_{PBS})}{(F_{TX} - F_{PBS})} \tag{3}$$

where $FI_{CL}$ is the fluorescence intensity after treatment with CL, $FI_{TX}$ corresponds to the positive control (100% leakage), and $FI_{PBS}$ corresponds to negative control (0% leakage). The quenching effect of TX on calcein was accounted for by measuring the fluorescence intensity of calcein in the presence and absence of TX and calculating a correction factor applied to $F_{TX}$:

$$\text{Correction Factor} = \frac{FI_{Calcein}}{FI_{Calcein+TX}} \tag{4}$$

The concentration of calcein was determined by measuring the fluorescence of calcein entrapped within the LUVs post purification. Samples were loaded into a black 96-well plate (Corning, Kennebunk, ME) and measured for two hours, with readings taken every minute, on a Cytation 5 plate reader (BioTek, Winooski, VT) using an excitation wavelength of 495 nm and an emission wavelength of 515 nm.

## Cell culture

Experiments were performed using the TR146 buccal squamous carcinoma cell line obtained from the European Collection of Authenticated Cell Cultures (ECACC 10032305) and grown in Dulbecco's Modified Eagle Medium (DMEM, Gibco) supplemented with 10% fetal bovine serum (FBS) and 1% penicillin-streptomycin. Cells were authenticated via STR profiling. Mycoplasma contamination was ruled out by PCR (Abcam 289834). All experiments were performed in serum-free DMEM.

## Western blot

TR146 cells were grown to confluency and starved overnight in serum-free DMEM. Cells were treated with 15 µM peptide for two hours. Post-incubation, cells were lysed on ice with TEN-T buffer (50 mM Tris-HCl pH 7.5, 100 mM NaCl, 1 mM EDTA, 1% Triton-X 100) containing phosphatase (Sigma) and protease (ThermoFisher) inhibitors for 30 min. Lysates were collected post centrifugation (20 min, 13,000 RPM, 4 °C) and separated on 10% SDS-PAGE gels before transfer to 0.45 µm nitrocellulose membranes. Blots were probed with primary antibodies for c-Fos (1:500, Cell Signaling Technology, 2250 S) and phospho-MKP1 (1:1000, Cell Signaling Technology, 2857 S) overnight. Fluorescent secondary antibodies (1:10,000, goat-anti mouse IRDye 680, goat-anti rabbit IRDye 800, LI-COR) were used for detection on a LI-COR Odyssey CLx. Human β-actin (1:1000, Abcam, ab6276) was used as a loading control and protein expression and phosphorylation was quantified using ImageStudio software.

## Acknowledgements

We thank Jennifer Schuster (University of Tennessee) for comments on the manuscript, and Ed Wright (University of Tennessee) for assistance with the AUC experiments. We are also thankful to Erwin London for scientific advice. This work was partially funded by awards NIH R35GM140846 and R01GM120642 (to F.N.B) and NSF 1709792 and 2122027 (to G.M.K.).

## Additional information

### Funding

| Funder | Grant reference number | Author |
| --- | --- | --- |
| National Institutes of Health | R35GM140846 | Francisco N Barrera |
| National Science Foundation | 1709792 | Gavin M King |
| National Institutes of Health | R01GM120642 | Francisco N Barrera |

| Funder | Grant reference number | Author |
| --- | --- | --- |
| National Science Foundation | 2122027 | Gavin M King |

The funders had no role in study design, data collection and interpretation, or the decision to submit the work for publication.

## Author contributions

Charles M Russell, Formal analysis, Investigation, Methodology, Writing – original draft, Writing – review and editing; Katherine G Schaefer, Conceptualization, Formal analysis, Investigation, Writing – original draft, Writing – review and editing; Andrew Dixson, Robert J Pyron, Daiane S Alves, Nicholas Moore, Elizabeth A Conley, Ryan J Schuck, Tommi A White, Investigation; Amber LH Gray, Investigation, Writing – original draft; Thanh D Do, Formal analysis, Investigation, Writing – original draft, Writing – review and editing; Gavin M King, Francisco N Barrera, Conceptualization, Formal analysis, Supervision, Funding acquisition, Investigation, Visualization, Methodology, Writing – original draft, Project administration, Writing – review and editing

## Author ORCIDs

Charles M Russell http://orcid.org/0000-0003-2489-657X
Katherine G Schaefer http://orcid.org/0000-0003-2180-799X
Amber LH Gray http://orcid.org/0000-0003-4126-868X
Daiane S Alves http://orcid.org/0000-0001-9154-4748
Thanh D Do http://orcid.org/0000-0002-1978-4365
Gavin M King http://orcid.org/0000-0002-5811-7012
Francisco N Barrera http://orcid.org/0000-0002-5200-7891

## Decision letter and Author response

Decision letter https://doi.org/10.7554/eLife.75490.sa1
Author response https://doi.org/10.7554/eLife.75490.sa2

# Additional files

## Supplementary files

- Transparent reporting form
- Supplementary file 1. Agilent 6560 IMS-QTOF parameters.

## Data availability

All data generated or analysed during this study are included in the manuscript and supporting file. Source data files have been provided for mass photometry data (Figures 1E and 6B).

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
