## [Editor Report]

This study by Russell et al. reveals interesting details of the mechanism of pore formation of a newly identified peptide toxin secreted by *Candida albicans* (candidalysin). Using atomic force microscopy along with other techniques, the authors demonstrate that pre-assembly of polymers in solution is the first step in the formation of permeabilizing pores, and identify an intriguing inactive mutant. This manuscript will be of interest to several fields, in particular to microbiologists and structural biologists studying pore forming proteins and peptides.

---

## [Decision Letter]

**Decision letter after peer review:**

Thank you for submitting your article "The *C. albicans* virulence factor Candidalysin polymerizes in solution to form membrane pores and damage epithelial cells" for consideration by *eLife*. Your article has been reviewed by 3 peer reviewers, and the evaluation has been overseen by a Reviewing Editor and Richard Aldrich as the Senior Editor. The following individual involved in review of your submission has agreed to reveal their identity: Ioan iacovache (Reviewer #2).

The reviewers all agreed that this is an interesting study that demonstrates a novel mechanism for membrane permeabilization by toxins, but found that some of the claims require additional experiments. These experiments are considered to be minimal, and already overlap with some of the studies already presented. Therefore, it is anticipated that they could be carried out in a reasonable amount of time and included in the revised manuscript. If this is not possible, then another option is to revise the manuscript significantly to tamper these claims. The following list outlines the essential revisions required:

Essential revisions:

1) While the data shows the existence of higher order assemblies in solution, the specific 8-mer octameric assembly does not appear to be well supported here. Further experiments are required to identify that the 8-mer exists in solution, to support the proposed mechanism that the 8-mer is the basic unit that seeds CL into larger structures. There are many experiments that may provide this information, such as cross-linking with SDS-PAGE, native PAGE or SEC with MALS. If these experiments are not possible or do not provide clear information to differentiate between alternate higher oligomeric states, which is admittedly difficult for an octamer species, then the paper still requires significant revisions to temper the claims that the mechanism relies on the octamer species, allowing for the possibility that other assemblies may be factor into the formation of the larger CL structures.

Additionally, the model shown in Figure 1D should be moved to a supplementary figure where a more complete description of the modeling variability and quality can be depicted. Alternate plausible models should be shown here to provide a more objective analysis of the possible oligomeric species.

2) The mutants studied here should be featured more completely with additional data characterizing their behavior, particularly SDS and native-PAGE, CD and oriented CD and solution oligomerization to determine that the effect observed is specific to the change in oligomerization.

3) Please explain the reasoning behind changing lipid compositions in various experiments. Is it possible that these membrane changes may impart differences on the observations? Please elaborate on this.

4) Please see and respond to recommended changes suggested by the reviewers below.

*Reviewer #1 (Recommendations for the authors):*

Specific queries/requests for revision to what is overall a fine manuscript, in my opinion:

– A really confusing aspect of the authors' interpretation is that they postulate a mechanism by which linear polymers of octamers can land on the membrane to form "rimmed pores" (see Figure 8). This raises the question of what defines the size of the polymers in solution. If these polymers are in equilibrium, the probability would maximum for either the shortest unit (an octomer) or to a maximum size, which in practice means that the size distribution would be determined by some kinetic trapping. In that case, the polymer size is expected to be strongly dependent on detailed experimental protocols and not be very robust. And I would expect that the size/area distribution for the "rimmed pores' would be much wider than for the "unrimmed pores", since the latter have a area that is (to within some flexibility) defined by the loop-shaped polymers in solution. Figure 3C shows some difference in area distributions, but it is far from clear how robust this areal analysis is given the difference in pore profiles. Remarkably, the AFM data in Figure 3A, B, D suggest that rimmed and unrimmed pores have very similar areas, suggesting that their size is set by a very similar assembly mechanism. The manuscript conclusions should be clarified, substantiated or adjusted in light to this. Another way of interpreting this is by noting that the assembly mechanism appears relatively robust, with the total pore assembly being rather robust against whether that assembly happens in solution or on the membrane.

*Reviewer #2 (Recommendations for the authors):*

While the majority of the presented data is clear and easily accessible several additions could benefit the reader:

1. The sequence of the peptide should be given in the text early on, currently is only shown in figure 6 and not well referenced in the text.

2. How the peptide was prepared (synthesis) is also not clearly mentioned in the text.

3. Figure 1A showing an SDS-PAGE of the peptide is not well explained. Is this a standard SDS-PAGE where the sample was boiled? In addition, a native PAGE could be shown to visualize other oligomeric species that are not resistant to SDS-PAGE.

4. In addition, the circular dichroism measurements of the bulk of peptide in solution is hard to interpret as a mixture of different species. If possible, one would expect that the oligomers and polymers could be analyzed separately (after separation by gel filtration or other techniques).

5. Figure 3 AB shows AFM images of rimmed and unrimmed pores in two separate panels. If one image could be shown with both types of pores it would strengthen the message, currently, at a glance at the figure one would assume they are different experimental conditions.

6. Negative stain EM of the polymers on membranes could be shown assuming that they are visible. Cryo-EM would be a better option to see polymers embedded in the membrane if the authors have access to a cryo-EM facility. This would strengthen the paper however cryo-EM could by itself be a follow-up paper.

7. Figure 4, solubilizing the bilayer with detergent is an elegant way to image the sample however is hard to rule out that the linear polymers observed are not circular oligomers that are disturbed by the detergent solubilization step or oligomers that polymerize in presence of detergent. In addition, the figure 5B could also show the CL in buffer plus detergent (or solubilized).

8. Both mutants should be featured more extensively in the paper, in particular SDS-PAGE (and native), oligomerization state, etc. According to the presented model, the G4W mutant should be able to oligomerize into the building block while losing the ability to form the longer polymers. If this is indeed the case it would help as both controls for the CD and oriented CD as well as for follow-up studies that could try to solve the structure of the building block.

9. Some of the techniques used in the paper could benefit from short introductions.

10. Some figures do not mention the scale bar size

*Reviewer #3 (Recommendations for the authors):*

Please find below comments to the paper and further suggestions to improve it.

1. Please include the whole name of the species in the title: *Candida albicans*.

2. I believe the name of the protein should be written as 'candidalysin' and not 'Candidalysin', as also used for other toxin names.

3. Results section starts that oligomerization of the peptide in solution was indicated by SDS-PAGE, but not much is commented further on the mass on the higher band (8-mer?) or other additional bands and how they were sure they belonged the same peptide (and no potential impurity). Or that if you see the higher band and that it belongs to CL oligomer, it must be a strong complex to resist SDS.

What would a simple size exclusion chromatography show? What would NATIVE-PAGE show (higher oligomers) and native-MS?

4. Data that lead to figure 1 show the probability of existence of 8-mers as a single unit. They do not suggest yet that they are ordered as described in the model in Figure 1D. This may be suggested with later results. Anyways, in the model, N- and C-terminus of the peptides should be marked so we can understand the orientation of the peptides in the proposed model of the 8-mer. Please comment on why CC-Type2-II model was used as a base for MD model of CL 8-mer.

5. Analytical UC results. 'The sedimentation velocity results reveal a low sedimentation peak, likely corresponding to a CL monomer (8), and several larger assemblies, in agreement with the IM-MS data analysis.'

Why is reference 8 here quoted? Please mark more exactly on Figure 1E what you describe in the text (what is monomer (1 peptide?), what is 8-mer etc). What exactly is meant by 'several larger assemblies, in agreement with the IM-MS data analysis'? Please described in a quantitative manner.

6. Figure 2A/B. Please use either expression 'loop' or 'ring', band not mixed.

7. Page 7 of combined PDF: Taken together, these results support the notion that CL assembles into an 8-mer in solution.

… and polymers of 8-mer, right?

8. Can you comment on possibility (or impossibility) of head-to-head dimerization of 8-mers?

9. It is rather hard to understand Figure 5A. Please, comment on this: 'Specifically, the modeled areas predict that unrimmed pores contain six to eight head-to-toe 8- mers, while rimmed pores are formed by six to twelve side-to-side 8-mers.'

How do you explain this difference between unrimmed and rimmed in terms of 8-mer copy No (6-8 and 6-12 8-mers, respectively)? If the unrimmed ends up as the rimmed and the rimmed could have more 8-mers than unrimmed?

10. Figure 5B. Which lipids were used? What type of lipid vesicles?

11. Figure 5B shows that CL forms, as expected (3), an a-helical structure in solution, as evidenced by the spectral minima at 208 and 222 nm, and that the presence of lipid vesicles promote modest additional helix formation.

Please comment 'additional helix formation', what could contribute to this: Elongation of the peptide helices (some of it is not structured as seen from the model)? More peptides included, more 8-mer subunits?

12. Model of the dimer of 8-mers, Suppl Figure 6. How was the model prepared? Please mark N and C termini. Is the depicted dimer head-to-toe? Hard to see.

13. Page 11 of the merged PDF: 'Our 8-mer model predicts that the Nt is slightly kinked (Figure 1D), and glycine residues confer flexibility to -helices (18).'

There is something missing in this sentence. Please rewrite. The first part is saying one thing (from the model) and the other part something else.

14. Also page 11: 'We reasoned that mutation of residue G4 (Figure 6A), located at the Nt kink, might hamper CL polymerization.'

Could it also hamper helix-helix interaction with 8-mer? Or when in the rimmed pore?

https://www.ncbi.nlm.nih.gov/pmc/articles/PMC1300449/pdf/10465772.pdf

15. Please provide CD spectra for G4W and I9A, to correct folding of mutants (and if comparable to WT)

16. Why in AFM DOPC was used and POPC for calcein release studies? And then POPC/POPE/cholesterol in OCD? Is there any lipid specificity?

Usage of lipids between different approaches should be (more) consistent.

17. Please provide info on orientation of the peptides (N and C termini). Add scale for AFM images.

In the figure legend: 'Upon insertion, pores exist in a dynamic equilibrium between unrimmed and rimmed pores, which cause cellular toxicity.'

Are only the rimmed pores functional (transmembrane)? Please state clearly.

Would you expect any conformational changes in the rimmed pore – in terms of elongation of helices in comparison to unrimmed (CD spectra?)

Can you comment on energy cost from going from unrimmed form to the rimmed form?

18. General question: what is more correct to say: unrimmed or non-rimmed?

19. Page 15 and general: Are the pores (unrimmed and rimmed) homogeneous in size? What would be the size in nm? Are they really as big as those of CDCs? To me, they look significantly smaller.

20. Monomeric forms of CDCs (perfringolysin) is 50-60 kDa, in the text it sounds as this was the Mw of the pore. First you compare the size of CL pores to CDC pores (huge) and in the next sentence you turn to hemolysins (which are 10x smaller than of CDCs and are completely different to CDC). It sounds confusing. Please revise this.The pores of CDCs and hemolysins are very different to those of CL, the mechanism of pore formation is very different (to the one proposed by you).

Hemolysins, CDC, MACPF etc undergo large conformational changes upon pore formation. No conformational changes in CL case? Within 8-mers? What about switching between the flat polymers to upward position of 8-mer (unrimmed vs rimmed)?

21. Page 15. Such structure is reminiscent of the vestibule of – hemolysin or the perimeter of pores formed by the N-terminal segment of gasdermin-D.

What did you mean here? Is there a vestibule in your pore? Your model of CL pores is completely different and can not be directly compared to hemolysin or gasdermin etc. Unless you have better structural data, e.i. cryoEM.

What do you mean by 'perimeter'? What is this in nm?

22. Page 16. The 8-mer might rotate because the three K residues located at the Ct … (Figure 6A) are expected to be interacting with lipid molecules in the rimmed pore configuration.

Please, elaborate on this? How exactly you propose for helices in 8-mer to be oriented (in either horizontal or vertical orientation).

23. Page 16. it. This behavior contrasts with the arcs formed by suilysin, as these rigid open polymers were able to remove lipid molecules from the bilayer to cause perforations (20).

Arcs in suilysin are completely different than in your case in terms os structure (built by protomers).

24. Methods should be described in the same order as presented in the Results section.

25. Peptide preparation: please avoid using statements such as 'were prepared as described previously (31, 32)'

Please provide a brief description how you did it.

All used program, chemicals and equipment should be referenced (paper or manufacturer etc) SEDFIT (?), carbon grids, protein standards (which ones).

26. Preparation of LUVS: what was the filter size?

27. OCD: The amount of lipid on each slide was 85.3 nmol,

Concentration of the lipid was calculated based on which lipid?

---

## [Author Response]

Essential revisions:1) While the data shows the existence of higher order assemblies in solution, the specific 8-mer octameric assembly does not appear to be well supported here. Further experiments are required to identify that the 8-mer exists in solution, to support the proposed mechanism that the 8-mer is the basic unit that seeds CL into larger structures. There are many experiments that may provide this information, such as cross-linking with SDS-PAGE, native PAGE or SEC with MALS. If these experiments are not possible or do not provide clear information to differentiate between alternate higher oligomeric states, which is admittedly difficult for an octamer species, then the paper still requires significant revisions to temper the claims that the mechanism relies on the octamer species, allowing for the possibility that other assemblies may be factor into the formation of the larger CL structures.

As the editor mentions, it is difficult to experimentally establish the stoichiometry of a high-order oligomer. We suspect that the additional data obtained using some of the experimental techniques suggested might not definitively support that the oligomer is indeed an 8-mer (and not, for instance, an 7-mer or a 9-mer). Therefore, we consider that the most prudent course is to follow the route suggested by the editor, and we have tempered our claim that the 8-mer definitively is the particles that assembles into a polymer. Accordingly, the updated manuscript has been modified to include this change throughout the text. Additionally, we have added a new closing section, titled “Limitations of the study”, where we discuss that we cannot rule out that the stoichiometry of the oligomer is slightly different, despite the presence of an 8-mer being supported by three different techniques (mass photometry, atomic force microscopy, and mass spectrometry). Instead of mentioning that the mechanism for polymer formation does require an 8-mer, we introduce the term “intermediate oligomer“, the presence of which is observed by AUC, AFM and TEM. We have additionally modified the abstract to point out that we merely propose that the basic structural unit in polymer formation is a CL 8-mer.

Additionally, the model shown in Figure 1D should be moved to a supplementary figure where a more complete description of the modeling variability and quality can be depicted. Alternate plausible models should be shown here to provide a more objective analysis of the possible oligomeric species.

As suggested, we have moved the 8-mer from Figure 1D into the new Figure 1- Supplementary 1. We also provide additional explanation of the interpretation of the IM-MS data a new Figure 1- Supplementary 2. We briefly summarize our major points here: given a nominal of n/z (oligomer size to charge ratio), one can determine n if z is known. The isotope spacings (mainly due to the difference between ^13^C and ^12^C) suggest that the octamer (Δm/z = 0.125) is among the major oligomers at m/z 3311. Because CD data showed that CL adopts a largely α-helical conformation in the absence of lipid, we modeled the 8-mer as a cylinder composed of 8 helices. The resulting theoretical collision cross section (CCS) agrees well with the experimental CCS if we assign the ATD feature at ~ 32 ms to be an octamer.

We followed the reviewers’ suggestion and we explicitly considered whether the feature could be assigned as a 9-mer. Based on structural similarity and polypeptide length, the closest 9-helix structure found corresponded to the central segment of the integral membrane light-harvesting complex II (LH2) (PDB ID 1NKZ; 2 Å resolution). The resulting 9-mer model of CL is shown In Author response image 1. We computed its theoretical CCS, and the value is significantly larger than our experimental value (if the feature at 32 ms were assigned as a 9-mer), i.e., 4,723 Å2 vs. 2,923 Å2. This result argues against the possibility of CL forming an 9-mer.

**Author response image 1. sa2fig1:** 

2) The mutants studied here should be featured more completely with additional data characterizing their behavior, particularly SDS and native-PAGE, CD and oriented CD and solution oligomerization to determine that the effect observed is specific to the change in oligomerization.

We have performed new experiments for the G4W and I9A mutants (shown in figure 5C and Figure 6—figure supplement 3), which support our previous conclusions. The new CD data show that WT and I9A have similar levels of helicity in buffer solution and in the presence of lipid vesicles. The loss-of-function mutant G4W showed lower levels of helicity. Oriented CD (OCD) results also showed agreement between WT and I9A, while G4W again showed changes; in this case, the OCD spectrum of G4W showed less transmembrane features compared to WT, as revealed by a ratio closer to 1 of the ellipticity at 208 and 225 nm. We attribute the decrease in overall OCD signal to the reduced level of overall helicity observed in the CD result of G4W. These results are now discussed into the manuscript.

We performed native-PAGE experiments, but this technique was not able to resolve any oligomers for CL, for reasons that we don’t fully understand.

3) Please explain the reasoning behind changing lipid compositions in various experiments. Is it possible that these membrane changes may impart differences on the observations? Please elaborate on this.

The reviewer is correct in thinking that changes in lipid composition could potentially affect the experimental results. To address this potential problem, we have repeated the OCD experiments. Initially these were performed using lipid bilayers composed of cholesterol, PE and PC lipids. The updated figure 5C shows data performed in POPC, which shows similar results. With regards to the rest of experiments, DOPC was used for the AFM experiments because we have had more success with this lipid spreading on the mica surface than with POPC. However, we do not anticipate qualitative changes in experimental results between DOPC and POPC bilayers. The new experiments allow for a better comparison between experimental results, as all were performed in lipids of identical headgroup composition.

Reviewer #1 (Recommendations for the authors):Specific queries/requests for revision to what is overall a fine manuscript, in my opinion:– A really confusing aspect of the authors' interpretation is that they postulate a mechanism by which linear polymers of octamers can land on the membrane to form "rimmed pores" (see Figure 8). This raises the question of what defines the size of the polymers in solution. If these polymers are in equilibrium, the probability would maximum for either the shortest unit (an octomer) or to a maximum size, which in practice means that the size distribution would be determined by some kinetic trapping. In that case, the polymer size is expected to be strongly dependent on detailed experimental protocols and not be very robust. And I would expect that the size/area distribution for the "rimmed pores' would be much wider than for the "unrimmed pores", since the latter have a area that is (to within some flexibility) defined by the loop-shaped polymers in solution. Figure 3C shows some difference in area distributions, but it is far from clear how robust this areal analysis is given the difference in pore profiles. Remarkably, the AFM data in Figure 3A, B, D suggest that rimmed and unrimmed pores have very similar areas, suggesting that their size is set by a very similar assembly mechanism. The manuscript conclusions should be clarified, substantiated or adjusted in light to this. Another way of interpreting this is by noting that the assembly mechanism appears relatively robust, with the total pore assembly being rather robust against whether that assembly happens in solution or on the membrane.

One of the conclusions put forth in the Discussion section is that the rimmed pores could arise from a conformational change of unrimmed pores. Thus, it is unsurprising that the areas of the rimmed and unrimmed pores are similar. Indeed, if our hypothesis is correct, we expect the area of the rimmed pores to be slightly smaller, due to asymmetry of the fundamental subunit. A slight area reduction is consistent with our AFM data comparing unrimmed and rimmed pores (Figure 3C). The kinetics of the loop assembly itself are intriguing and under investigation, but outside the scope of this work. However, we have added a new section at the end of the discussion where we mention that the loops that we observed might not be in equilibrium, and hence are under kinetic control. We stand behind the robustness of our comparison between WT and the two mutants, as experiments were performed under identical conditions.

Reviewer #2 (Recommendations for the authors):While the majority of the presented data is clear and easily accessible several additions could benefit the reader:1. The sequence of the peptide should be given in the text early on, currently is only shown in figure 6 and not well referenced in the text.

The sequence of WT CL has been introduced into the manuscript, at the beginning of the Methods section.

2. How the peptide was prepared (synthesis) is also not clearly mentioned in the text.

Details on the synthesis of the peptides can be found in the Peptide Preparation section of the Methods.

3. Figure 1A showing an SDS-PAGE of the peptide is not well explained. Is this a standard SDS-PAGE where the sample was boiled? In addition, a native PAGE could be shown to visualize other oligomeric species that are not resistant to SDS-PAGE.

We have introduced additional detail on how the SDS-PAGE experiment was performed. The legend of Figure 1 now describes that the samples were not boiled. We appreciate the suggestion of the reviewer to perform native PAGE experiments. We have tried for a long time to carry out these experiments, but in our hands, they did show a single band for CL; therefore, this technique apparently cannot be used to study CL oligomerization.

4. In addition, the circular dichroism measurements of the bulk of peptide in solution is hard to interpret as a mixture of different species. If possible, one would expect that the oligomers and polymers could be analyzed separately (after separation by gel filtration or other techniques).

We followed the suggestion of the reviewer, and performed gel filtration chromatography to separate different oligomeric species of CL. However, the results were not satisfactory. When we performed the experiment in standard buffer, with 150 mM NaCl, we observed no peaks at all, suggesting unspecific interactions of the peptide with the column matrix. Performing FPLC experiments at lower ionic strength did not yield robust results either. It therefore does not seem that gel filtration can be used to isolate the CL oligomers observed in solution.

5. Figure 3 AB shows AFM images of rimmed and unrimmed pores in two separate panels. If one image could be shown with both types of pores it would strengthen the message, currently, at a glance at the figure one would assume they are different experimental conditions.

We have included a new supplementary figure for this instance (Figure 3 —figure supplement 3) that shows rimmed and unrimmed pores in the same image.

6. Negative stain EM of the polymers on membranes could be shown assuming that they are visible. Cryo-EM would be a better option to see polymers embedded in the membrane if the authors have access to a cryo-EM facility. This would strengthen the paper however cryo-EM could by itself be a follow-up paper.

This work lays a foundation for future studies. We agree with the reviewer that employing cryo-EM on the CL membrane system would make a very interesting follow up investigation.

7. Figure 4, solubilizing the bilayer with detergent is an elegant way to image the sample however is hard to rule out that the linear polymers observed are not circular oligomers that are disturbed by the detergent solubilization step or oligomers that polymerize in presence of detergent. In addition, the figure 5B could also show the CL in buffer plus detergent (or solubilized).

We thank the reviewer for this observation. Quantification of the area of the polymers showed no significant changes in the polymers caused by the detergent. Based on the reviewer comment, we tested for any potential detergent disturbance by quantification of a second parameter, the fraction of polymers that close into a loop. A table has been added to Figure 4—figure supplement 1 displaying the results of three independent experiments. These results suggest that indeed the detergent might make the small fraction of CL loops even smaller. However, this effect would not change any of the main conclusions derived from this AFM assay, as we also report data where bilayer removal occurred spontaneously (in the absence of detergent).

8. Both mutants should be featured more extensively in the paper, in particular SDS-PAGE (and native), oligomerization state, etc. According to the presented model, the G4W mutant should be able to oligomerize into the building block while losing the ability to form the longer polymers. If this is indeed the case it would help as both controls for the CD and oriented CD as well as for follow-up studies that could try to solve the structure of the building block.

We have performed additional experiments of the two mutants, consisting of CD and oriented CD. The new data are presented in Figure 5C and Figure 6-Suppl 3. We also performed native PAGE experiments, but the technique did not yield relevant information as it only showed a single band.

9. Some of the techniques used in the paper could benefit from short introductions.

To provide the reader with methodology background we have added several additional references, and included an explanatory sentence on the mass photometry technique (page 5).

10. Some figures do not mention the scale bar size.

We thank the reviewer for pointing out this oversight. Scale bar sizes have been added to Figure 8, which were missing.

Reviewer #3 (Recommendations for the authors):Please find below comments to the paper and further suggestions to improve it.1. Please include the whole name of the species in the title: Candida albicans.

The manuscript title has been changed accordingly.

2. I believe the name of the protein should be written as 'candidalysin' and not 'Candidalysin', as also used for other toxin names.

The capitalization has been changed throughout the manuscript, as suggested.

3. Results section starts that oligomerization of the peptide in solution was indicated by SDS-PAGE, but not much is commented further on the mass on the higher band (8-mer?) or other additional bands and how they were sure they belonged the same peptide (and no potential impurity). Or that if you see the higher band and that it belongs to CL oligomer, it must be a strong complex to resist SDS.What would a simple size exclusion chromatography show? What would NATIVE-PAGE show (higher oligomers) and native-MS?

The purity of the CL is higher than 95%, as determined by HPLC, ruling out that any of the highly populated species observed in the SDS-PAGE or AUC data correspond to impurities. SDS-resistant oligomers are not at all uncommon for helical membrane proteins that form oligomers (e.g., the KcsA potassium channel; https://pubmed.ncbi.nlm.nih.gov/16245951/).

We followed the reviewer suggestion and performed several iterations of FPLC and native-PAGE experiments with CL. However, the results were inconclusive; no peak at all was observed in FPLC experiments when experiments were performed with physiological salt concentration, as recommended per the column instructions, suggesting artifactual binding between the peptide and the material of the column. When we performed Blue Native PAGE experiments with candidalysin, we could not observe any bands corresponding to oligomers, for reasons that we don’t currently understand.

4. Data that lead to figure 1 show the probability of existence of 8-mers as a single unit. They do not suggest yet that they are ordered as described in the model in Figure 1D. This may be suggested with later results. Anyways, in the model, N- and C-terminus of the peptides should be marked so we can understand the orientation of the peptides in the proposed model of the 8-mer. Please comment on why CC-Type2-II model was used as a base for MD model of CL 8-mer.

As suggested by the editor, the 8-meric model has been moved to the Figure 1 —figure supplement 1, where we now identify the N- and C-termini. We thank the reviewer for giving us a chance to clarify the construction of the octamer model. In the new Figure 1 —figure supplement 2, we show the isotope spacing roughly corresponding to z = +8, indicating that the most abundant species is n/z = 8/8 (an octamer with +8 charges). Our assignments were consistent with the ion mobilities of species with the same number of charges but smaller in size (n/z = 7/8) and species with the same size but higher charge state (n/z = 8/9). We modeled the octamer by taking into account two pieces of evidence: (a) the circular dichroism data showed that CL has α-helix structure, and (b) the homology modeling using ROSETTA that also suggested that the peptide monomer is α-helical. Thus, we start with an α-helix monomeric unit and construct an octamer. To do so, we searched the RCSB databank for α-helix multimers. The hits included 1PI7, 2H6Y, 3R3K, 6O35, 6G56, and 6G67. We then used EMBOSS to align the primary sequence of CL with the primary sequences of the structures from the RCSB. The best alignment was obtained with 6G67 which is a parallel eight-helix coiled coil CC-Type2-II. We then threaded the primary sequence of CL onto the backbone of 6G67 and minimized the structure with molecular dynamics (MD) using the Gromacs software and the CHARMM force field. Finally, we calculated the theoretical collision cross-section (CCS) of the octamer model and compared it with the experimental CCS. It appears that the theoretical CCS (in helium; 2500 Å2) is in an excellent agreement with the experimental CCS (2539 Å2). We used the Mobcal software and the trajectory method to compute the theoretical CCS.

The primary sequence alignments of CL and candidate peptides from the RCSB databank are shown in Author response image 2.

5. Analytical UC results. 'The sedimentation velocity results reveal a low sedimentation peak, likely corresponding to a CL monomer (8), and several larger assemblies, in agreement with the IM-MS data analysis.'Why is reference 8 here quoted? Please mark more exactly on Figure 1E what you describe in the text (what is monomer (1 peptide?), what is 8-mer etc). What exactly is meant by 'several larger assemblies, in agreement with the IM-MS data analysis'? Please described in a quantitative manner.

The citation of reference 8 has been removed, as it was out of place. We purposely avoided to try to correlate the AUC peaks observed at 2 S, 4.5 S and 9 S with specific oligomeric states, as we suspect that the different oligomeric states of CL are dynamically interacting in the time resolution of the AUC experiments. In interacting systems, the boundaries are “reaction” boundaries and not species (Ann N Y Acad Sci; 1969 Nov 7;164(1):192-225. doi: 10.1111/j.1749-6632.1969.tb14039.x.). According to this, unless you have a kinetically frozen system, the sedimentation coefficients cannot be accurately explained as species, as they merely represent the extent of reaction. Therefore, to avoid an erroneous interpretation, we think it is safer to acknowledge that AUC might not be able to quantitatively identify the identity of the oligomers we observe.

6. Figure 2A/B. Please use either expression 'loop' or 'ring', band not mixed.

We thank the reviewer for identifying this oversight. Figure 2 has been adjusted accordingly.

7. Page 7 of combined PDF: Taken together, these results support the notion that CL assembles into an 8-mer in solution.… and polymers of 8-mer, right?

That is correct, but we found it was clearer to instead make this point at the end of the next short paragraph.

8. Can you comment on possibility (or impossibility) of head-to-head dimerization of 8-mers?

We are happy to speculate! One can certainly contemplate the possibility of head-to-head dimers. However, in this situation polymer growth would require two types of interactions, head-to-head and tail-to-tail, instead of one. However, these two interaction modes would have to overcome electrostatic repulsion, as both the Nt and Ct would be charged at neutral pH. We are of the philosophical opinion that Nature is more likely to find the easiest solution, i.e. only head-to-tail. Despite any hypotheses that we might have, the fact is that we don’t have data to support either orientation possibility for 8-mer dimerization. We indicate this in the updated version of the manuscript (in the results and the new Limitations of the Study s in the Discussion).

9. It is rather hard to understand Figure 5A. Please, comment on this: 'Specifically, the modeled areas predict that unrimmed pores contain six to eight head-to-toe 8- mers, while rimmed pores are formed by six to twelve side-to-side 8-mers.'How do you explain this difference between unrimmed and rimmed in terms of 8-mer copy No (6-8 and 6-12 8-mers, respectively)? If the unrimmed ends up as the rimmed and the rimmed could have more 8-mers than unrimmed?

Discussion of the correlation between the inner area of the simulated loops and the pores has been updated. To guide the reader, a new cartoon is included in Figure 5A, which directly shows the measurements being compared.

10. Figure 5B. Which lipids were used? What type of lipid vesicles?

As described in the methods, we used LUVs for the experiment. The lipid was POPC.

11. Figure 5B shows that CL forms, as expected (3), an a-helical structure in solution, as evidenced by the spectral minima at 208 and 222 nm, and that the presence of lipid vesicles promote modest additional helix formation.Please comment 'additional helix formation', what could contribute to this: Elongation of the peptide helices (some of it is not structured as seen from the model)? More peptides included, more 8-mer subunits?

We have introduced in the manuscript that it is possible that the additional helical formation might result from an elongation of the helix.

12. Model of the dimer of 8-mers, Suppl Figure 6. How was the model prepared? Please mark N and C termini. Is the depicted dimer head-to-toe? Hard to see.

In the Molecular Dynamics Simulation section, we describe how the 16-mer model was obtained. Briefly, starting from the octamer, we manually align the two octamer units in different orientations (head-to-head, head-to-toe, toe-to-toe) and then performed 45-ns simulations in explicit water using the GROMACS software and the CHARMM force field. The resulting structure of the head-to-toe, which is the most stable, is shown below. We have added a Figure 6—figure supplement 1, which contains the detailed figure of two 8-mers interacting in a head-to-toe orientation, where the N-termini are highlighted in red, and the C-termini in blue.

More importantly, we noted that in the IM-MS data of m/z 3311, there is another major feature with shorter arrival time than the octamer. Based on the difference in arrival time between the two features, the left feature should be a larger oligomer than the octamer (in IM-MS, higher-order oligomers travel faster than smaller one due to the effect of charges outweighs the effect of size). Now, if we assign the feature to a 16-mer, the experimental CCS is 4446 Å2. We computed the theoretical CCS of the 16-mer model from MD and the theoretical CCS is 4517 Å2.

13. Page 11 of the merged PDF: 'Our 8-mer model predicts that the Nt is slightly kinked (Figure 1D), and glycine residues confer flexibility to -helices (18).'There is something missing in this sentence. Please rewrite. The first part is saying one thing (from the model) and the other part something else.

We have tightened the language of the sentence.

14. Also page 11: 'We reasoned that mutation of residue G4 (Figure 6A), located at the Nt kink, might hamper CL polymerization.'Could it also hamper helix-helix interaction with 8-mer? Or when in the rimmed pore?https://www.ncbi.nlm.nih.gov/pmc/articles/PMC1300449/pdf/10465772.pdf

The new language of this sentence includes the possibility of disturbance of the 8-mer.

15. Please provide CD spectra for G4W and I9A, to correct folding of mutants (and if comparable to WT)

These data are now provided, as a new Figure 6—figure supplement 3.

16. Why in AFM DOPC was used and POPC for calcein release studies? And then POPC/POPE/cholesterol in OCD? Is there any lipid specificity?Usage of lipids between different approaches should be (more) consistent.

As described above, we have repeated the OCD experiments in the lipid DOPC to increase lipid consistency. These results are shown on Figure 5C.

17. Please provide info on orientation of the peptides (N and C termini). Add scale for AFM images.In the figure legend: 'Upon insertion, pores exist in a dynamic equilibrium between unrimmed and rimmed pores, which cause cellular toxicity.'Are only the rimmed pores functional (transmembrane)? Please state clearly.Would you expect any conformational changes in the rimmed pore – in terms of elongation of helices in comparison to unrimmed (CD spectra?)Can you comment on energy cost from going from unrimmed form to the rimmed form?

We have introduced into the manuscript further discussion with regards to the relative toxicity of rimmed and unrimmed pores. As with regards of a helical extension being part of the conformational change, it is certainly an interesting possibility, but out data cannot resolve it. We think that both types of pores would be functional (able to permeate membranes), but the rimmed pore would probably be slightly more stable. Transmembrane pores are difficult to identify in AFM due to tip size constraints. This is mentioned on page 7 of the manuscript. A study of the energetic landscape of the peptides in the lipid bilayer would be very interesting. Such studies are challenging (usually involving force spectroscopy) and beyond the scope of this work. Any comments on the energetics of the conformational change undergone by the CL assemblies cannot be made at this point based on the experiments performed.

18. General question: what is more correct to say: unrimmed or non-rimmed?

Good point! We think that both options are similarly correct. We prefer “unrimmed”, as we find it easier to pronounce than the alternative proposed.

19. Page 15 and general: Are the pores (unrimmed and rimmed) homogeneous in size? What would be the size in nm? Are they really as big as those of CDCs? To me, they look significantly smaller.

As shown by the area histograms (Figure 3C), the unrimmed pores are slightly larger than the rimmed pores and have a broader distribution of areas. In terms of overall size, CL pores indeed exhibit diameters <80 Å, whereas CDCs are much bigger (~300 Å, https://doi.org/10.1016/j.bbamem.2015.11.017). The discussion has been modified to more precisely talk about pore dimensions.

20. Monomeric forms of CDCs (perfringolysin) is 50-60 kDa, in the text it sounds as this was the Mw of the pore. First you compare the size of CL pores to CDC pores (huge) and in the next sentence you turn to hemolysins (which are 10x smaller than of CDCs and are completely different to CDC). It sounds confusing. Please revise this.The pores of CDCs and hemolysins are very different to those of CL, the mechanism of pore formation is very different (to the one proposed by you).Hemolysins, CDC, MACPF etc undergo large conformational changes upon pore formation. No conformational changes in CL case? Within 8-mers? What about switching between the flat polymers to upward position of 8-mer (unrimmed vs rimmed)?

The relevant section of the manuscript has been clarified. We emphasize (*i*) that it is the *difference* between CL and these other peptides or protein systems that we wish to highlight and (*ii*) the CL mechanism is novel to our knowledge. We define that a small conformational switch, in the form of a conformational swell, occurs when rimmed pores are formed.

21. Page 15. Such structure is reminiscent of the vestibule of – hemolysin or the perimeter of pores formed by the N-terminal segment of gasdermin-D.What did you mean here? Is there a vestibule in your pore? Your model of CL pores is completely different and can not be directly compared to hemolysin or gasdermin etc. Unless you have better structural data, e.i. cryoEM.What do you mean by 'perimeter'? What is this in nm?

We have modified the text to clarify that CL pores do not contain a vestibule. We also removed the comment on the perimeter, as apparently it was not clear.

22. Page 16. The 8-mer might rotate because the three K residues located at the Ct … (Figure 6A) are expected to be interacting with lipid molecules in the rimmed pore configuration.Please, elaborate on this? How exactly you propose for helices in 8-mer to be oriented (in either horizontal or vertical orientation).

This sentence has been expanded in the manuscript to clarify this point.

23. Page 16. it. This behavior contrasts with the arcs formed by suilysin, as these rigid open polymers were able to remove lipid molecules from the bilayer to cause perforations (20).Arcs in suilysin are completely different than in your case in terms os structure (built by protomers).

We agree with the reviewer. The mechanism of CL pore formation appears to be distinct from that of the suilysin arcs that have been reported previously. The discussion has been revised accordingly.

24. Methods should be described in the same order as presented in the Results section.

As suggested, the Methods section has been re-ordered to agree with how the experiments were presented in the Results section.

25. Peptide preparation: please avoid using statements such as 'were prepared as described previously (31, 32)'Please provide a brief description how you did it.All used program, chemicals and equipment should be referenced (paper or manufacturer etc) SEDFIT (?), carbon grids, protein standards (which ones).

This point has been fixed.

26. Preparation of LUVS: what was the filter size?

A new table is included in the manuscript, *per eLife* requirements, containing this information.

27. OCD: The amount of lipid on each slide was 85.3 nmol,Concentration of the lipid was calculated based on which lipid?

The number of moles refers to total amount of lipid, and the concentration was calculated for the average molecular weight of the three lipids.